# Super-Linear: A Lightweight Pretrained Mixture of Linear Experts for Time Series Forecasting

**Liran Nochumsohn**  *lirannoc@post.bgu.ac.il*
*Faculty of Computer and Information Science, Ben-Gurion University*

**Raz Marshanski**  *razmar@post.bgu.ac.il*
*Faculty of Computer and Information Science, Ben-Gurion University*

**Hedi Zisling**  *hediz@post.bgu.ac.il*
*Faculty of Computer and Information Science, Ben-Gurion University*

**Omri Azencot**  *azencot@bgu.ac.il*
*Faculty of Computer and Information Science, Ben-Gurion University*

**Reviewed on OpenReview:** *https://openreview.net/forum?id=av8niDGYMk*

## Abstract

Time series forecasting (TSF) is critical in domains like energy, finance, healthcare, and logistics, requiring models that generalize across diverse datasets. Large pre-trained models such as Chronos and Time-MoE show strong zero-shot (ZS) performance but suffer from high computational costs. In this work, we introduce Super-Linear, a lightweight and scalable mixture-of-experts (MoE) model for general forecasting. It replaces deep architectures with simple frequency-specialized linear experts. A lightweight spectral gating mechanism dynamically selects relevant experts, enabling efficient, accurate forecasting. Crucially, resampling during training exposes the model to diverse frequency regimes, while a flexible input adaptation strategy allows it to handle varying inference lengths. Despite its simplicity, Super-Linear demonstrates strong performance across benchmarks, while substantially improving efficiency, robustness to sampling rates, and interpretability. The implementation of Super-Linear is publicly available at: https://github.com/azencot-group/SuperLinear

## 1 Introduction

Time series forecasting plays a vital role in many high-impact domains, from optimizing energy grids and managing financial risk to supporting clinical decision-making and supply chain operations Taylor & Letham (2018); Salinas et al. (2020); Lim et al. (2021). As these systems increasingly rely on automated forecasting tools, the demand grows for models that can generalize across a wide range of temporal settings without manual tuning. However, many real-world scenarios involve small or heterogeneous datasets with unique periodic patterns, limited labels, and shifting dynamics Norton et al. (2025). In such environments, training specialized models per dataset, often referred to as full-shot forecasting (FSF), is computationally costly and often impractical, as it requires extensive retraining on the complete historical data of every new target domain. This motivates the need for *zero-shot forecasting* (ZSF), a setting in which a single model can generalize to entirely unseen datasets and domains without any retraining or fine-tuning Shi et al. (2024); Liu et al. (2024d;c); Das et al. (2024).

Recent work has made significant progress toward this goal. Large pretrained models such as Timer-XL, TimeMoE, and TimesFM (Liu et al., 2024c; Shi et al., 2024; Das et al., 2024) employ transformer-based backbones, input patching, and masked pretraining to achieve strong zero-shot performance across domains. However, these gains come at a high computational cost. For instance, TimeMoE was trained over several

days using 28 A100-80GB GPUs, whereas Chronos can take over 4 seconds to generate an output for a batch size of 64 as presented in Tab. 3, and Fig. 1 for a general overview. Additionally, the complexity of these architectures also poses a challenge for interpretability, as their sophisticated designs make it difficult to understand or explain forecasting decisions. Developing models that combine strong generalization with greater transparency still remains a key challenge, which often hinders accessibility and practical deployment. As zero-shot forecasting becomes more practical, it is imperative to develop models that preserve strong generalization while dramatically reducing the computational and memory footprint.

Recent findings show that carefully constructed linear models can outperform deep transformer-based architectures across several TSF benchmarks (Zeng et al., 2023; Toner & Darlow, 2024; Ni et al., 2024; Nochumsohn et al., 202t). Consequently, linear forecasting models offer a promising foundation for efficient ZSF and *full-shot forecasting* (FSF). They are fast, lightweight, and well-aligned with the temporal structure of many natural sequences. However, a single linear model typically lacks the expressivity needed to generalize across diverse datasets (Nochumsohn et al., 202t). Capturing multiple overlapping frequencies with one parameter set induces what we term *frequency confusion*, where different temporal patterns compete for representation and degrade accuracy. This limitation is especially pronounced in zero-shot settings, where effective generalization requires both robustness to distribution shifts and adaptability to novel temporal patterns. Taken together, these challenges may explain why linear models remain relatively underexplored in zero-shot forecasting within pre-trained frameworks.

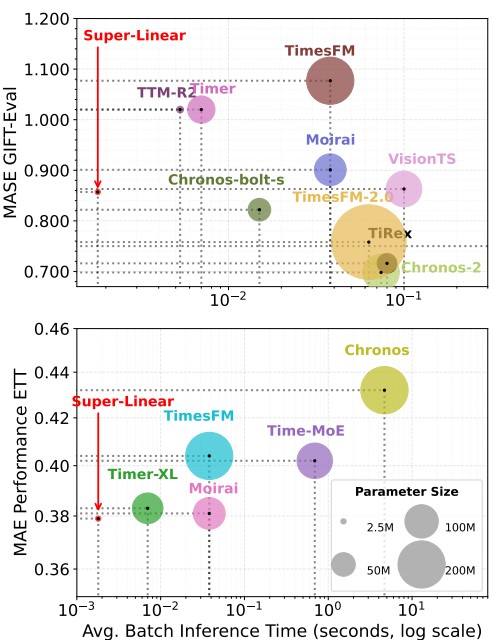

Toward bridging this gap, we introduce *Super-Linear*, a sparse mixture-of-experts (MoE) model that brings modularity and frequency-awareness to linear forecasting. Super-Linear comprises of linear heads most of which are trained to specialize in a particular frequency regime, using resampled data to ensure abundance and diversity of periodic signals. A lightweight, interpretable gating mechanism routes each input sequence to a subset of experts based on its spectral signature, dynamically assembling predictions without shared weights. Complementary experts and simple heuristics (e.g., naïve repeat, mean

Figure 1: Performance versus inference time trade-off across different prominent pre-trained TSFM on the GIFT-Eval and LTSF benchmarks.

forecasting) provide further flexibility, allowing for the recovery of accurate forecasts even in ambiguous or low-frequency settings. Crucially, our approach emphasizes frequency-aware data construction, where resampling is used not only for augmentation but as a primary mechanism to expose the model to diverse temporal patterns during inference, which we find to be essential for strong performance.

Super-Linear combines the scalability and interpretability of linear models with the adaptability of MoE architectures. It is trained using standard procedures, requires no auxiliary loss functions or sophisticated optimization schemes, and runs efficiently on a single GPU. As we show in Fig. 1, Super-Linear achieves competitive performance relative to state-of-the-art zero-shot and full-shot forecasting models, while offering substantial improvements in inference speed and model size. We evaluate Super-Linear against leading state-of-the-art baselines across both zero-shot and full-shot regimes. Despite its small size, just 2.5 million parameters, Super-Linear achieves substantial improvements in zero-shot forecasting, reducing average MSE by up to 26.2% compared to prominent models like Chronos and TimesFM. It also outperforms Timer-XL in both MSE and MAE on more than twice as many benchmarks, while being only 3% of its size. In the GIFT-Eval benchamrk (Aksu et al., 2024), Super-Linear achieves a competitive MASE score surpassing the performance of TSFM such as VisionTS , Moirai, and TTM (Chen et al., 2024; Woo et al., 2024; Ekambaram et al., 2024).

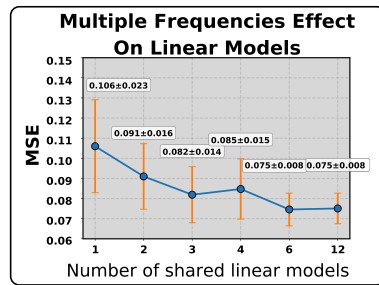 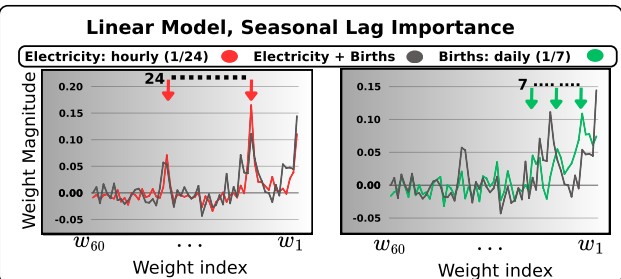

Figure 2: Left: Forecasting performance of linear models on 12 sine-wave datasets with varying frequencies and added random walk noise. Performance improves progressively with more experts. Right: Weight sensitivity to seasonal lags—training on datasets with different seasonality (e.g., Births vs. Electricity) leads to divergent weight structures, suboptimal when shared.

## 2 Related Work

**Pre-trained Model in TSF.** Recent advancements in pretrained models for vision and text (Zhou et al., 2024) have accelerated the development of large time series foundation models for zero-shot, few-shot, and full-shot forecasting (Liang et al., 2024). Modern models like UniTime (Liu et al., 2024b), TEMPO (Cao et al., 2023), and GPT4TS (Zhou et al., 2023) adopted GPT-based architectures, inspiring works such as MOMENT (Goswami et al., 2024) and TimesFM (Das et al., 2024), which integrate transformers (Vaswani et al., 2017), input patching, and masked training to handle diverse inputs. Chronos (Ansari et al., 2024) leverages a T5-based architecture (Raffel et al., 2020) with discrete time tokenization, while Moirai (Woo et al., 2024) employs an encoder-only transformer with multi-patch size projections and binary attention biases for universal forecasting. Timer and Timer-XL (Liu et al., 2024d;c), as decoder-only generative models, use next-token prediction for flexible time series processing. Finally, Time-MoE (Shi et al., 2024), a decoder-only mixture-of-experts model, effectively scales parameters, enhancing accuracy and efficiency in forecasting.

**Linear and Sparse TSF Models.** Before deep time series forecasting (TSF) gained traction, statistical and linear models such as ARIMA, Exponential Smoothing, and least squares regression (Shumway & Stoffer, 2000) dominated TSF. In recent years, MLP (multilayer perceptron) and transformer-based models initially outperformed these methods with models such as Informer, Autoformer, FEDformer, and N-HiTS (Zhou et al., 2021; Wu et al., 2021; Zhou et al., 2022; Challu et al., 2023), however, recent research has demonstrated that linear and sparse models such as DLinear and RLinear (Zeng et al., 2023; Toner & Darlow, 2024; Li et al., 2023) can be competitive or even superior. This has sparked interest in sparse and frequency-aware TSF models, including FITS and SparseTSF (Xu et al., 2023; Lin et al., 2024b), which leverage the frequency domain for improved forecasting. MTLinear (Nochumsohn et al., 202t) addresses gradient conflicts with variate cross-correlation, enhancing robustness against transformers with linear models, while CycleNet (Lin et al., 2024a) employs a shallow MLP with learnable recurrent cycles to model periodic patterns efficiently. These approaches provide scalable and interpretable forecasting solutions, particularly for resource-constrained applications.

**Mixture of Experts.** Mixture of Experts (MoE) (Cai et al., 2024; Jacobs et al., 1991) enhances efficiency and scalability by dynamically selecting specialized sub-models for different tasks, reducing computational overhead while maintaining high performance. Although widely used in language models (Cai et al., 2024), MoE has been sparingly applied to time series forecasting, beginning with the Mixture of Experts Model (MEM) (Zeevi et al., 1996), which extends autoregressive models into a nonlinear universal approximator. Models such as FEDformer (Zhou et al., 2022) utilize MoE for different average pooling selection, while Moirai-MoE and Time-MoE (Woo et al., 2024; Shi et al., 2024), incorporate MoE within their designs. MoLE (Ni et al., 2024) introduces weighted linear experts for in-domain forecasting, and FreqMoE (Liu, 2025) processes distinct frequency bands for prediction. Distinguishably, our approach scales these ideas into a centralized, linear-enhanced pretrained model capable of handling diverse datasets, while maintaining a

sparse architecture built around a single linear-expert MoE layer, unlike other MoE methods that rely on intermediate, non-linear modules.

## 3 Super-Linear Mixture-of-Experts

**Problem Formulation.** Given a historical sequence of observations $X_{1:L} = [x_1, x_2, \ldots, x_L]^T \in \mathbb{R}^L$, where $L$ represents the number of lookback time steps, the goal in time series forecasting is to predict the future $H$ steps $Y_{1:H} = X_{L+1:L+H} = [x_{L+1}, \ldots, x_{L+H}]^T \in \mathbb{R}^H$. Super-Linear MoE leverages a modular architecture combining linear maps and expert gating mechanisms to enhance forecasting accuracy across diverse horizons with a single model, ranging from short-term (e.g., one step) to long-term (e.g., 720 steps and beyond). By dynamically selecting relevant experts based on input characteristics, the model adapts to varying frequencies and amplitudes. Furthermore, the model design employs channel-wise independence (univariate) to ensure scalability to high-dimensional datasets, enabling Super-Linear to generalize effectively across multivariate forecasting tasks in real-world scenarios.

In this work, we evaluate Super-Linear under both zero-shot (ZS) and in-domain (full-shot, FS) forecasting settings. In our usage, zero-shot denotes applying the pretrained model to entirely unseen datasets with no additional training or adaptation, whereas in-domain forecasting refers to performance after training the model directly on the target dataset.

**Motivation.** Time series is naturally periodic, yet common methods, such as seasonal-trend decomposition or covariate timestamp embeddings (Ni et al., 2024), struggle with scalability and mixed-frequency signals (Theodosiou, 2011; Woo et al., 2022; Qin et al., 2024). Inspired by Fourier analysis (Shumway & Stoffer, 2000), we view a time series as a weighted sum of its dominant periodic components, with $\hat{\omega}_i$ denoting the characteristic frequencies and $\alpha_i$ are their amplitudes,

$$X \approx \alpha_1 X_{\hat{\omega}_1} + \alpha_2 X_{\hat{\omega}_2} + \cdots + \alpha_k X_{\hat{\omega}_k} \ . \tag{1}$$

Linear forecasting models that take the form of an autoregressive process remain surprisingly effective Zeng et al. (2023), and often learn to emphasize seasonal lags—e.g., daily or weekly offsets—when such structure exists. However, capturing multiple overlapping frequencies with a single weight vector is fundamentally limited. As different seasonalities compete, shared weights induce "frequency confusion" and degrade performance (see Fig. 2). To address this, we propose approximating forecasts using a mixture of frequency-specialized linear experts. Given input $X \in \mathbb{R}^{1 \times L}$ and output $Y \in \mathbb{R}^{1 \times H}$, where $L$ and $H$ are the lookback and horizon, we have:

$$Y \approx \hat{Y} = \beta_1 F_{\hat{\omega}_1}(X) + \beta_2 F_{\hat{\omega}_2}(X) + \cdots + \beta_k F_{\hat{\omega}_k}(X) \ , \tag{2}$$

where each $F_{\hat{\omega}_i}$ models a specific frequency and $\beta_i$ are data-dependent gating weights. This modular design mitigates interference, supports dynamic frequency selection, and improves generalization in complex forecasting tasks. A schematic illustration of our architecture is presented in Fig. 3.

**Model Overview.** Let $X \in \mathbb{R}^{1 \times L}$ and $Y \in \mathbb{R}^{1 \times H}$. Super-Linear produces a forecast $\hat{Y} \in \mathbb{R}^{1 \times H}$ over a horizon of $H$ steps. This forecast is computed as a weighted sum over $N$ experts, each of which contributes a prediction $F_i(X) \in \mathbb{R}^{1 \times H}$, modulated by a gating weight $G_i(X) \in \mathbb{R}$:

$$\hat{Y} = \sum_{i=1}^{N} G_i(X) F_i(X) \ , \tag{3}$$

**Linear Experts.** Each expert $F_i$ is a learnable or non-learnable linear model. The learnable experts consist of $N_f$ frequency-specific modules and $N_c$ complementary ones, implemented via RLinear—a linear layer combined with instance normalization using RevIN Toner & Darlow (2024). This normalization mitigates distribution shifts by reapplying the input's original scale after transformation. The output of each learnable expert is:

$$F_i(X) = \text{RevIN}^{-1} \left( \text{RevIN}(X) W_i + b_i \right) \ , \tag{4}$$

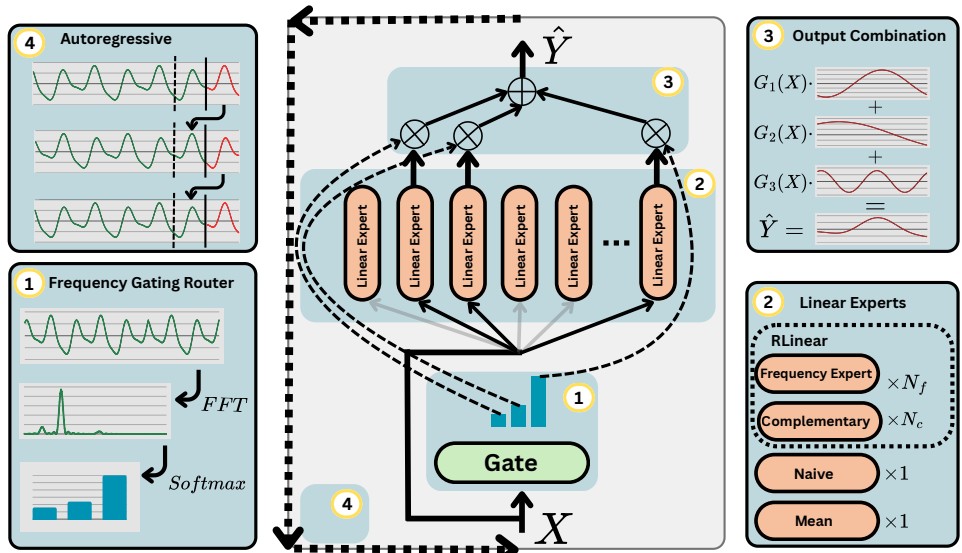

Figure 3: Super-Linear architecture overview. A frequency-aware gating router computes sparse scores from the input frequencies, dynamically selecting a subset of linear experts, (1) including linear experts (2) whose predictions are combined to produce the final forecast (3).

where $W_i \in \mathbb{R}^{L \times H}$ and $b_i \in \mathbb{R}^H$ are trainable. Frequency experts are pre-trained with data associated with a specific frequency $\hat{\omega}_i$, covering diverse periodicities, while complementary experts are trained jointly to capture residual temporal signals. More on the frequencies, and how they were obtained, is given in App. D.3). Two non-learnable experts, the *naïve* (last value) and *mean* (input average), serve as robust heuristics in low-frequency or short-sequence regimes. Including them improves performance across benchmarks (Tab. 2).

**Gating Function.** The weights $G_i(X)$ for combining expert outputs are derived through a sparse gating mechanism (Shazeer et al., 2017) that dynamically selects the most relevant experts based on the input's spectral properties. Specifically, we compute a gating matrix $g(X) \in \mathbb{R}^{1 \times N}$, where each entry reflects the affinity between the input and a given expert. These entries are computed using the normalized periodogram $I(X - \bar{X})$ (Shumway & Stoffer, 2000; Schuster, 1898) of the centered zero-mean input. Formally,

$$g(X) = \frac{I(X-\bar{X})}{\|I(X-\bar{X})\|_1} W_g + b_g \ , \ I(X - \bar{X}) = \left| \frac{\text{FFT}(X-\bar{X})}{\sqrt{2M}} \right|^2 ,$$

where $W_g \in \mathbb{R}^{M \times N}$ and $b_g \in \mathbb{R}^N$ are trainable parameters, FFT denotes the Fast Fourier Transform, and $2M$ is the number of spectral bins. Since the input is real-valued, the spectrum is symmetric and only the first $M$ coefficients are retained. $I(X - \bar{X})$ is then normalized using the $l_1$ norm.

**Sparse Gating.** To encourage sparsity and specialization, only the top-$k$ scoring experts are activated for each input. During training, we inject Gaussian noise $\mathcal{R}^{\text{noise}} \in \mathbb{R}^{1 \times N}$ into the scores to promote exploration. The final gating weights, $G(X) \in \mathbb{R}^{1 \times N}$, are computed using a softmax applied only over the top-$k$ scores per column, i.e., TopK eliminates the $(N - k)$ smallest elements of $g(X) + \sigma \mathcal{R}^{\text{noise}}$ for $\sigma = 0.1$:

$$G(X) = \text{softmax}\left(\text{TopK}(g(X) + \sigma \mathcal{R}^{\text{noise}}, k)\right) \ , \tag{5}$$

$$\text{TopK}_i(H, k) = \begin{cases} H_i, & \text{if } i \in \text{top-}k, \\ -\infty, & \text{otherwise.} \end{cases} \tag{6}$$

With $H_i$ denoting the $i$-column of $H$. By incorporating spectral structure into the gating process and combining this with a diverse set of linear experts, Super-Linear is able to efficiently adapt to a variety of temporal phenomena, balancing model simplicity with strong empirical performance.

**Training Mechanism.** Super-Linear adopts a two-stage training framework, illustrated in Fig. 4: *Stage 1: Frequency expert pre-training.* Each expert is trained independently on data aligned with its designated fundamental frequency $\hat{\omega}_i$. To expose each expert to diverse temporal patterns, we apply extensive resampling augmentations using linear interpolation. Unlike prior work focused on dataset diversity, our emphasis is on *frequency abundance*, capturing a broad range of frequency instances rather than dataset heterogeneity. We thus select a subset of datasets from Monash Godahewa et al. (2021) and PEMS Liu et al. (2022) that provide sufficient sequence length for downsampling, specifically, datasets with a total sequence length above $5,000$ timesteps. Full details on dataset selection and resampling are provided in App. C.

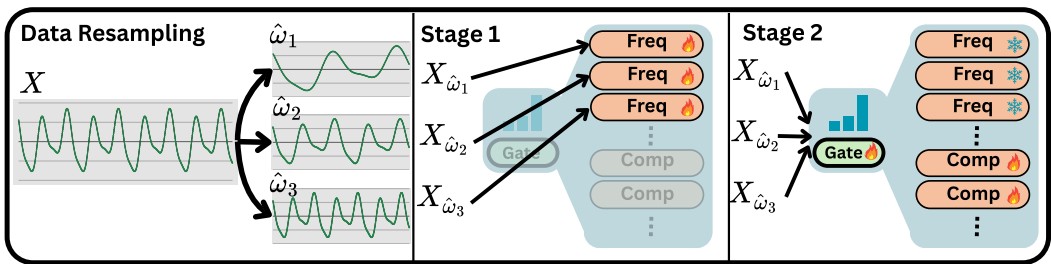

Figure 4: Super-Linear training framework. Data is resampled to enrich frequency diversity. Stage 1: Each expert is trained independently on a predefined frequency $\omega_i$. Stage 2: The router and complementary layers are trained with frozen experts to enable dynamic expert selection.

*Stage 2: MoE router training.* After pre-training, we freeze the frequency expert weights and proceed to train the MoE router jointly with the complementary expert layers. During this stage, only the router (gating mechanism) and the lightweight complementary layers are updated. These complementary layers are specifically designed to refine predictions by capturing residual patterns not handled by the frequency experts. The router learns to dynamically select the most suitable expert based on the input signal characteristics. This stage supports either in-domain downstream tasks or pre-training on multi-frequency datasets to enable zero-shot generalization.

**Varying Lookbacks Adaptability.** Although Super-Linear is a linear forecaster with fixed lookback weights, it employs a heuristic to handle variable input lengths. If the lookback $L_{\text{input}}$ is shorter than the training length ($L_{\text{train}} = 512$), the series is upsampled via linear interpolation; if longer, it is downsampled while preserving signal energy. After inference, outputs are rescaled to the original resolution. While this procedure implicitly alters signal frequencies, Super-Linear remains robust to such variations. Consequently, the model can operate with any input length, as long as the input contains more than one data point. More details and a detailed algorithm are provided in Appendix B.9.

## 4 Experiments

### 4.1 Experimental Details

**Pre-training Datasets and Resampling.** This work emphasizes frequency diversity over sheer dataset volume. For pre-training, we use two widely adopted sources: Monash (Godahewa et al., 2021) and PEMS (Liu et al., 2022). From Monash, only datasets with more than 5,000 samples are included, ensuring sufficient data for resampling. Resampling, both upsampling and downsampling data augmentation, is performed on-the-fly during training via linear interpolation, eliminating the need to store multiple copies and saving space. We use the frequencies detailed in App. D.3. The same pre-training data is used across both training stages, as outlined in Sec. 3.

**Training and Implementation.** All Super-Linear models are trained on a single NVIDIA RTX3090 (24GB) using PyTorch (Paszke et al., 2019). In Stage 1, linear experts are trained independently and later combined into the Super-Linear architecture. We use the Adam optimizer with MSE loss and without

auxiliary loss factors. For full-shot, the top-k experts are chosen via validation loss. All Super-Linear models use a lookback window of 512 and forecast a horizon of 96; longer horizons are predicted autoregressively. Full hyperparameters are detailed in App. F.

**Baselines and Evaluations.** We compare our method to top-performing models. For zero-shot, we benchmark against Timer-XL, ROSE, Time-MoE, Moirai, TimesFM variants, Chronos variants, Sundial, and TiRex (Liu et al., 2024c; Wang et al.; Shi et al., 2024; Woo et al., 2024; Das et al., 2024; Ansari et al., 2024; Liu et al., 2025b; Auer et al., 2025), and focus mostly on models available on GitHub or Hugging Face. Evaluation follows the LTSF benchmark in both zero-shot and full-shot setups, using ETTm1, ETTm2, ETTh1, ETTh2, Electricity, Traffic, and Weather datasets Liu et al. (2024c); Shi et al. (2024), including the GIFT-Eval benchmark (Aksu et al., 2024), which covers a wide range of dataset domains and sampling rates. The appendix includes additional evaluations such as full-shot (see Tab. 4), and extra model comparisons.

**GIFT-Eval: Model Performance and Model Size.** GIFT-Eval (Aksu et al., 2024) is a comprehensive benchmark for time series forecasting, designed to fairly assess the zero-shot and universal forecasting capabilities of foundation models across diverse domains, frequencies, prediction lengths, and variates. We evaluate Super-Linear against state-of-the-art models including Sundial, VisionTS Liu et al. (2025b); Chen et al. (2024), Chronos, and TimesFM Ansari et al. (2024); Das et al. (2024) in Fig. 5. Despite having the smallest parameter count (**2.5M**), Super-Linear achieves competitive results, delivering a **16%** MASE reduction over TTM, the next smallest model, while being competitive or surpassing

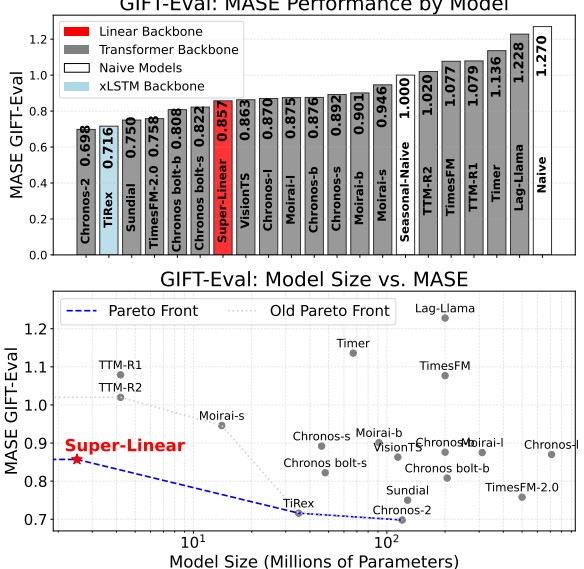

Figure 5: GIFT-Eval performance and parameter count of Super-Linear compared to prominent foundation models in TSF . The MASE score represents the geometric mean MASE across datasets, normalized by the seasonal-naive. Last update: 01/03/2026.

larger Transformer-based models such as VisionTS, TimesFM, and Sundial. Note that the GIFT-Eval leaderboard is continuously updated, and our results reflect peer-reviewed models available at the time stated in Fig. 5.

## 4.2 Main Results

**LTSF: Zero-shot Forecasting and Model Size.** In this section, we present the zero-shot (ZS) performance of Super-Linear in comparison to state-of-the-art (SOTA) models, given in Tab. 1. Despite having a relatively small number of parameters, Super-Linear achieves average mean squared error (MSE) reductions of **26.2%**, **14.2%**, **8.5%**, **7.1%**, and **6.3%** compared to Chronos, TimesFM, Moirai, Time-MoE Large, and Time-MoE Base, respectively. Additionally, Super-Linear outperforms Timer-XL by achieving **17** best scores in MSE and **19** best scores in mean absolute error (MAE), compared to Timer-XL's 4 and 0 best scores. Remarkably, Super-Linear accomplishes this while being only **3%** the size of Timer-XL, with only **2.5M** parameters compared to 84M. To ensure fairness, we compare against models with similar lookback windows, excluding evaluations that rely on excessively long lookbacks, parameter search, or ensembles.

**Zero-Shot Generalization Across Varying Sampling Rates.** Existing pre-trained models exhibit a strong reliance on hourly-sampled datasets, which comprise between 59.4% and 93.1% of their training data (Fig. 6). In contrast, Super-Linear promotes frequency diversity by training on a balanced set of predefined sampling rates using linear interpolation. To assess robustness to underrepresented frequencies, an area often

Table 1: Zero-shot forecasting results (MSE & MAE) and model size. A dash (–) indicates datasets included in the model's pre-training or no reported results. **Red** and blue highlight the best and second-best results per row, respectively. Time-MoE results are from (Shi et al., 2024); others from (Liu et al., 2024c).

| Dataset | Horizon | Super-Linear MSE | MAE | Timer-XL MSE | MAE | ROSE MSE | MAE | Time-MoE base MSE | MAE | Time-MoE Large MSE | MAE | Moirai MSE | MAE | TimesFM MSE | MAE | Chronos MSE | MAE |
|---|---|---|---|---|---|---|---|---|---|---|---|---|---|---|---|---|---|
| ETTh1 | 96 | 0.369 | 0.392 | 0.369 | 0.391 | 0.382 | 0.408 | 0.357 | **0.381** | **0.350** | 0.382 | 0.376 | 0.392 | 0.414 | 0.404 | 0.440 | 0.393 |
| | 192 | 0.399 | 0.410 | 0.405 | 0.413 | 0.400 | 0.420 | **0.384** | **0.404** | 0.388 | 0.412 | 0.412 | 0.413 | 0.465 | 0.434 | 0.492 | 0.426 |
| | 336 | 0.416 | **0.421** | 0.418 | 0.423 | **0.404** | 0.426 | 0.411 | 0.434 | 0.411 | 0.430 | 0.433 | 0.428 | 0.503 | 0.456 | 0.550 | 0.462 |
| | 720 | 0.424 | **0.439** | 0.423 | 0.441 | **0.420** | 0.447 | 0.449 | 0.477 | 0.427 | 0.455 | 0.447 | 0.444 | 0.511 | 0.481 | 0.882 | 0.591 |
| | Avg. | 0.402 | **0.416** | 0.404 | 0.417 | 0.401 | 0.425 | 0.400 | 0.424 | **0.394** | 0.420 | 0.417 | 0.419 | 0.473 | 0.444 | 0.591 | 0.468 |
| ETTh2 | 96 | **0.279** | 0.342 | 0.283 | 0.342 | 0.298 | 0.362 | 0.305 | 0.359 | 0.302 | 0.354 | 0.294 | **0.330** | 0.315 | 0.349 | 0.308 | 0.343 |
| | 192 | **0.333** | **0.379** | 0.340 | 0.379 | 0.336 | 0.385 | 0.351 | 0.386 | 0.364 | 0.385 | 0.365 | 0.375 | 0.388 | 0.395 | 0.384 | 0.392 |
| | 336 | 0.354 | **0.398** | 0.366 | 0.400 | **0.353** | 0.399 | 0.391 | 0.418 | 0.417 | 0.425 | 0.376 | 0.390 | 0.422 | 0.427 | 0.429 | 0.430 |
| | 720 | **0.389** | **0.425** | 0.397 | 0.431 | 0.395 | 0.432 | 0.419 | 0.454 | 0.537 | 0.496 | 0.416 | 0.433 | 0.443 | 0.454 | 0.501 | 0.477 |
| | Avg. | **0.339** | 0.386 | 0.347 | 0.388 | 0.346 | 0.394 | 0.366 | 0.404 | 0.405 | 0.415 | 0.363 | **0.382** | 0.392 | 0.406 | 0.406 | 0.410 |
| ETTm1 | 96 | 0.317 | **0.352** | 0.317 | 0.356 | 0.512 | 0.460 | 0.338 | 0.368 | **0.309** | 0.357 | 0.363 | 0.356 | 0.361 | 0.370 | 0.454 | 0.408 |
| | 192 | 0.357 | **0.375** | 0.358 | 0.381 | 0.512 | 0.462 | 0.353 | 0.388 | **0.346** | 0.381 | 0.388 | 0.375 | 0.414 | 0.405 | 0.567 | 0.477 |
| | 336 | 0.393 | 0.395 | 0.386 | 0.401 | 0.523 | 0.470 | 0.381 | 0.413 | **0.373** | 0.408 | 0.416 | **0.392** | 0.445 | 0.429 | 0.662 | 0.525 |
| | 720 | 0.459 | 0.432 | **0.430** | 0.431 | 0.552 | 0.490 | 0.504 | 0.493 | 0.475 | 0.477 | 0.460 | **0.418** | 0.512 | 0.471 | 0.900 | 0.591 |
| | Avg. | 0.382 | 0.388 | **0.373** | 0.392 | 0.525 | 0.470 | 0.394 | 0.416 | 0.376 | 0.406 | 0.407 | **0.385** | 0.433 | 0.419 | 0.646 | 0.500 |
| ETTm2 | 96 | **0.179** | **0.265** | 0.189 | 0.277 | 0.224 | 0.309 | 0.201 | 0.291 | 0.197 | 0.286 | 0.205 | 0.273 | 0.202 | 0.270 | 0.199 | 0.274 |
| | 192 | **0.240** | **0.305** | 0.241 | 0.315 | 0.266 | 0.333 | 0.258 | 0.334 | 0.250 | 0.322 | 0.289 | 0.316 | 0.261 | 0.322 | 0.261 | 0.322 |
| | 336 | 0.293 | 0.339 | **0.286** | 0.348 | 0.310 | 0.358 | 0.324 | 0.373 | 0.337 | 0.375 | 0.329 | 0.350 | 0.360 | 0.366 | 0.326 | 0.366 |
| | 720 | 0.382 | **0.392** | **0.375** | 0.402 | 0.395 | 0.407 | 0.488 | 0.464 | 0.480 | 0.461 | 0.437 | 0.411 | 0.462 | 0.430 | 0.455 | 0.439 |
| | Avg. | **0.273** | **0.325** | 0.273 | 0.336 | 0.299 | 0.352 | 0.318 | 0.366 | 0.316 | 0.361 | 0.312 | 0.337 | 0.328 | 0.347 | 0.310 | 0.350 |
| Electricity | 96 | **0.141** | 0.239 | **0.141** | 0.237 | 0.209 | 0.307 | - | - | - | - | 0.160 | 0.250 | - | - | 0.154 | **0.231** |
| | 192 | **0.157** | **0.253** | 0.159 | 0.254 | 0.219 | 0.315 | - | - | - | - | 0.175 | 0.263 | - | - | 0.179 | 0.254 |
| | 336 | **0.175** | **0.270** | 0.177 | 0.272 | 0.236 | 0.330 | - | - | - | - | 0.187 | 0.277 | - | - | 0.214 | 0.284 |
| | 720 | **0.217** | **0.306** | 0.219 | 0.308 | 0.273 | 0.328 | - | - | - | - | 0.228 | 0.309 | - | - | 0.311 | 0.346 |
| | Avg. | **0.172** | **0.267** | 0.174 | 0.268 | 0.234 | 0.320 | - | - | - | - | 0.188 | 0.275 | - | - | 0.214 | 0.279 |
| Traffic | 96 | **0.414** | **0.296** | - | - | 0.572 | 0.407 | - | - | - | - | - | - | - | - | 0.562 | 0.378 |
| | 192 | **0.430** | **0.300** | - | - | 0.575 | 0.406 | - | - | - | - | - | - | - | - | 0.579 | 0.412 |
| | 336 | **0.449** | **0.309** | - | - | 0.588 | 0.411 | - | - | - | - | - | - | - | - | 0.594 | 0.420 |
| | 720 | **0.551** | **0.344** | - | - | 0.618 | 0.422 | - | - | - | - | - | - | - | - | 0.723 | 0.472 |
| | Avg. | **0.461** | **0.312** | - | - | 0.588 | 0.412 | - | - | - | - | - | - | - | - | 0.614 | 0.420 |
| Weather | 96 | **0.159** | **0.211** | 0.171 | 0.225 | 0.200 | 0.260 | 0.160 | 0.214 | **0.159** | 0.213 | 0.220 | 0.217 | - | - | 0.203 | 0.238 |
| | 192 | **0.207** | **0.254** | 0.221 | 0.271 | 0.239 | 0.288 | 0.210 | 0.260 | 0.215 | 0.266 | 0.271 | 0.259 | - | - | 0.256 | 0.290 |
| | 336 | **0.261** | **0.293** | 0.274 | 0.311 | 0.279 | 0.315 | 0.274 | 0.309 | 0.291 | 0.322 | 0.286 | 0.297 | - | - | 0.314 | 0.336 |
| | 720 | **0.333** | **0.341** | 0.356 | 0.370 | 0.340 | 0.357 | 0.418 | 0.405 | 0.415 | 0.400 | 0.373 | 0.354 | - | - | 0.397 | 0.396 |
| | Avg. | **0.240** | **0.275** | 0.256 | 0.294 | 0.264 | 0.305 | 0.266 | 0.297 | 0.270 | 0.300 | 0.288 | 0.282 | - | - | 0.292 | 0.315 |
| 1st Count | | **17** | **19** | 4 | 0 | 3 | 0 | 1 | 2 | 5 | 0 | 0 | 6 | 0 | 0 | 0 | 1 |
| Model Size | | **2.5M** | | 84M | | 7.4M | | 113M | | 453M | | 91M | | 200M | | 201M | |

overlooked in TSF, we evaluate performance across a range of resampled datasets (Tab. 2). Super-Linear achieves the best results in 8 out of 10 scenarios and demonstrates the most consistent performance overall, which we attribute to its frequency-aware training and linear expert design. In this setting, we follow the standard LTSF training, validation, and test protocol used for ETTm. We resample ETTm to match the corresponding sampling rate of 10, 20, 30, 40, and 50 minutes.

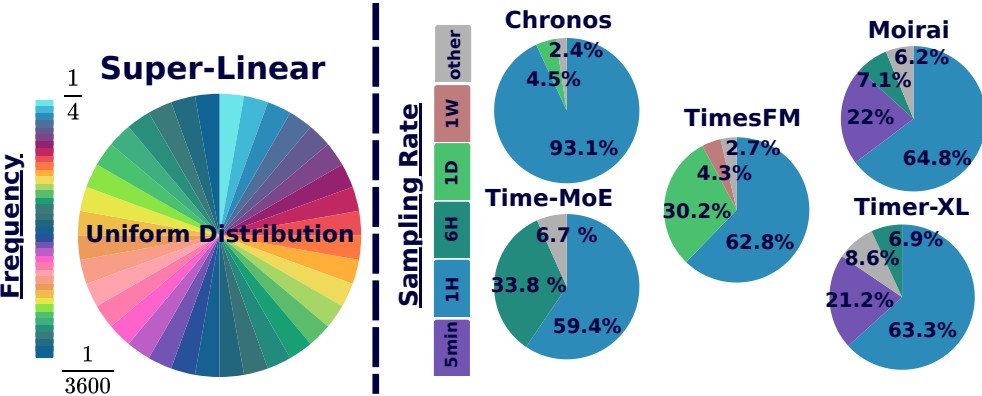

Figure 6: Dataset pre-training distribution for sampling rate or frequency. It is shown that other models rely heavily on hourly-sampled data. The frequencies for Super-Linear are given in App. 11.

Table 2: Left: Ablation study of Super-Linear by removing key components: long lookback inference downsampling (WO Inf Downsamp), complementary experts (WO Comp), expert types (WO Mean Naive), two-stage training (One Stage), and frequency resampling during training (WO Tr Resamp). Right: Zero-shot performance under varying sampling rates, highlighting robustness to changes in temporal resolution.

| Model | 10min ETTm1 | 10min ETTm2 | 20min ETTm1 | 20min ETTm2 | 30min ETTm1 | 30min ETTm2 | 40min ETTm1 | 40min ETTm2 | 50min ETTm1 | 50min ETTm2 |
|---|---|---|---|---|---|---|---|---|---|---|
| Super-Linear | **0.361** | **0.147** | **0.371** | **0.170** | **0.408** | **0.189** | **0.416** | **0.198** | 0.839 | 0.255 |
| TimesFM-2.0 | 0.443 | 0.155 | 0.503 | 0.198 | 0.447 | 0.201 | 0.562 | 0.222 | 0.589 | 0.239 |
| TimeMoE | 0.616 | 0.197 | 0.663 | 0.223 | 0.449 | 0.220 | 0.618 | 0.269 | 0.657 | 0.281 |
| TiRex | 0.544 | 0.168 | 0.453 | 0.182 | 0.429 | 0.194 | 0.464 | 0.216 | 0.463 | 0.231 |
| Moirai | 1.226 | 0.212 | 0.991 | 0.231 | 1.027 | 0.241 | 1.060 | 0.261 | 1.013 | 0.269 |
| Timer-XL | 0.609 | 0.161 | 0.841 | 0.199 | 0.437 | 0.208 | 0.868 | 0.234 | 0.865 | 0.252 |

| Method | Full | WO Inf Downsamp | WO Compl | WO Mean Naive | One Stage | WO Tr Resamp |
|---|---|---|---|---|---|---|
| MSE-LTSF | **0.265** | - | 0.268 | 0.268 | 0.282 | 0.423 |
| MAE-LTSF | **0.3** | - | 0.304 | 0.304 | 0.315 | 0.37 |
| GIFT-Eval MASE | **0.857** | 0.92 | **0.857** | 0.865 | 0.913 | 1.603 |

Figure 7: Top-$k$ expert behavior in the Super-Linear model. **Left:** Performance comparison across varying top-$k$ expert settings, with blue and red vertical lines indicating the $k$ values used during pretraining and inference, respectively. **Right:** Distribution of selected experts across different datasets using the pretrained model.

**Method Ablation.** Table 2 presents the ablation study, reporting average MSE and MAE on the LTSF benchmark and the geometric mean on GIFT-EVAL. The full Super-Linear model achieves the best overall performance. Removing long lookback downsampling for inference (WO Inf Downsamp) substantially reduces generalization on GIFT-EVAL ($0.857 \rightarrow 0.92$), though performance remains competitive with models such as TTM and Timer. Excluding complementary experts produces minor degradations (WO Compl), while removing the mean and naive experts (WO Mean Naive) has a more pronounced negative impact on GIFT-EVAL. Replacing the two-stage training with a single-stage setup (One Stage) leads to a clear performance drop, underscoring the importance of the proposed training scheme.

Most notably, both input construction and data augmentation emerge as the dominant factors affecting performance. Removing inference-time downsampling significantly harms generalization, highlighting the sensitivity of the model to input length adaptation. Even more critically, training only on the raw datasets without resampling (WO Tr Resamp) leads to the largest degradation across all benchmarks (e.g., GIFT-EVAL MASE $0.857 \rightarrow 1.603$), confirming that frequency-aware resampling is essential for exposing the model to diverse temporal patterns. These findings suggest that data construction—particularly resampling and input adaptation—has a greater impact on performance than most architectural components.

For the LTSF results, all experiments use a forecast horizon of 96, with a fixed lookback length of 512, making long lookback resampling inapplicable. Accordingly, the WO Inf Downsamp ablation is only evaluated on GIFT-EVAL, as LTSF benchmarks use a fixed lookback across all methods.

**Model Interpretability.** One of Super-Linear's key strengths lies in its interpretability through expert selection. Unlike deep MoE models such as Time-MoE, where multi-layered complexity hinders transparent analysis, Super-Linear employs a single expert layer, enabling direct examination of the softmax outputs. In Fig. 7, we present the learned expert distributions for ETT, Weather, and Electricity. For example, the Electricity dataset, which is sampled hourly, predominantly activates the 1/24 expert, corresponding to hour/day periodicity. The Weather dataset tends to favor the 1/144 expert (10 min/day), and also often

selects the naïve expert, likely indicating the presence of noise or low-frequency dynamics. These distributions expose clear links between sampling rates, seasonal patterns, and expert activation, offering valuable insights into Super-Linear's decision-making behavior.

**Top-k Experts Ablation.** Another insightful analysis involves evaluating performance as a function of the top-$k$ experts hyperparameter for the sparse MoE. We conduct experiments with varying $k \in 3, 6, 9, 12, 15$, as shown in Fig. 7. A key feature of Super-Linear is its ability to adapt the expert selection at inference time by recalibrating the expert distribution for any chosen $k \leq N$. Namely, we can pretrain on one $k$ and evaluate on another $k$. In this experiment, we assess the performance of a Super-Linear model pretrained with $k = 12$, comparing both inference-time adjustments (red curve) and retraining with different $k$ values (blue curve). Results show dataset-dependent behavior, with a preference for the retrained setup for ETTh1, ETTm2, and Weather; however, adjusting the top-$k$ at inference can be advantageous in some inference-time cases—for example, on ETTh1 and ETTm1, selecting $k = 3$ outperforms the original $k = 12$ configuration, despite the model being trained with $k = 12$. This highlights the practicality and adaptability of Super-Linear for zero-shot forecasting tasks.

## 4.3 Inference Time

In this section, we extend the discussion and summarize the results presented in Figure 1. Across models drawn from both benchmarks, Table 3 demonstrates that Super-Linear achieves substantially faster inference, delivering 8.3× speedup over Chronos-Bolt-small and 383.3× over TimeMoE, while maintaining competitive accuracy. These results highlight that Super-Linear provides strong inference time performance while maintaining competitive accuracy, despite its very small parameter count. The inference time represents the average time, in seconds, required to process a batch of size 64, lookback 512, and horizon 32.

Table 3: Inference time comparison of prominent TSFMs. Each result represents the average inference time with lookback 512, horizon 32, and batch size 64.

| Model | Size (M) | Time (sec) | Speedup |
|---|---|---|---|
| Super-Linear | **2.5** | **0.0018** | ×1 |
| TTM-R2-Zeroshot | 4.2 | 0.0053 | ×2.9 |
| Timer-XL | 84 | 0.0070 | ×3.9 |
| Chronos-Bolt-Small | 48 | 0.0150 | ×8.3 |
| Chronos-Bolt-Base | 205 | 0.0280 | ×15.5 |
| Moirai-base | 91 | 0.0380 | ×21.1 |
| TimesFM | 200 | 0.0380 | ×21.1 |
| TimesFM-2.0 | 500 | 0.0630 | ×35.0 |
| Chronos-2 | 120 | 0.0740 | ×41.1 |
| TiRex | 35 | 0.0800 | ×44.4 |
| VisionTS | 114 | 0.1000 | ×55.5 |
| Sundial | 128 | 0.4400 | ×244.4 |
| TimeMoE | 113 | 0.6900 | ×383.3 |
| Chronos-base | 201 | 4.6800 | ×2600.0 |

## 5 Theoretical Analysis

We provide a theoretical analysis of the gating mechanism through the lens of the *bias-complexity tradeoff* (Shalev-Shwartz & Ben-David, 2014), yielding bounds on the frequency approximation error. This framework helps assess the strengths and limitations of our frequency-specialized approach. Specifically, it decomposes the error of the ERM algorithm into a frequency-focused approximation error $\mathcal{E}_{\mathrm{app}}$, representing bias from limited model capacity, and estimation error $\mathcal{E}_{\mathrm{est}}$, arising from overfitting due to model complexity. By analyzing this tradeoff, we obtain theoretical insights into the robustness of the model and the gating mechanism. Assuming

$X$ and $Y(X)$ share the same periodicity, formally: $I^X = I^Y = I$, and for some $\gamma > 0$ we then get,

$$\|Y(X) - \hat{Y}(X)\| = \mathcal{E}_{\text{app}} + \mathcal{E}_{\text{est}} \tag{7}$$
$$\leq \sqrt{E_\perp} + \gamma \|X\|_2 \|\beta - G\|_1,$$

- $E_\perp$ bounds the *(frequency)-approximation error*: the signal energy outside the span of the $N$ expert frequencies.

- The second term, which bounds the *estimation error*, penalises how far the learned gating $G$ is from the optimal $\beta$. If training makes $\|\beta - G\|_1 \to 0$, the total error collapses to the intrinsic residual $\sqrt{E_\perp}$.

To ease notation, $G$, $\beta$, $E_\perp$ represent $G(X)$, $\beta(X)$, and $E_\perp(X)$, respectively. Adding more frequency and complementary experts reduces approximation error but risks overfitting, while too few experts lead to underfitting, reflecting the bias–complexity trade-off (Shalev-Shwartz & Ben-David, 2014). Incorporating natural frequencies D.3 introduces a meaningful inductive bias, ensuring sufficient representative experts to cover diverse datasets while controlling complexity to prevent overfitting. To further validate this analysis, in App. B.5, we empirically validate our inductive bias premise, showing that selecting frequencies at random degrades performance due to increased approximation error. In B.11.4, we provide the proof and provide an analysis with a stricter assumption on the shared periodicity.

## 6 Conclusion

Super-Linear is a sparse mixture-of-experts framework for zero-shot and full-shot time series forecasting that combines frequency-specialized linear experts with a spectral routing function to achieve efficient, accurate, and interpretable generalization across diverse benchmarks. Importantly, our findings highlight that data construction—particularly resampling and input adaptation during inference—plays a central role in enabling this performance, often outweighing architectural modifications. While effective and scalable, the reliance on linear experts limits performance on highly nonlinear or chaotic series, suggesting future extensions with nonlinear or hybrid experts and more advanced routing strategies to enhance adaptability in complex settings.

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

# A   Appendix

This appendix provides comprehensive supplementary material to support the main findings of our work. We present additional experimental results, ablation studies, theoretical analyses, and detailed descriptions of datasets and benchmark models. Section A covers full-shot and zero-shot forecasting results, including extensive comparisons with state-of-the-art baselines and ablation studies on model components and input lookback lengths. Section B offers a theoretical analysis of the Super-Linear model's error decomposition. Section C details the datasets used for both pre-training and evaluation, while Section D describes the benchmark models considered in our comparisons. Finally, we provide further insights into the frequency expert weights and hyperparameter configurations used in our experiments.

Table 4: Full-shot (in-domain) forecasting results. **Red** and blue highlight the best and second-best results per row, respectively. Super-Linear utilizes a lookback of 512.

| Dataset | Horizon | Super-Linear MSE | Super-Linear MAE | Timer-XL MSE | Timer-XL MAE | CycleNet MSE | CycleNet MAE | UniTST MSE | UniTST MAE | iTransformer MSE | iTransformer MAE | PatchTST MSE | PatchTST MAE | GPT4TS MSE | GPT4TS MAE | TimesNet MSE | TimesNet MAE | DUET MSE | DUET MAE | FEDformer MSE | FEDformer MAE | Autoformer MSE | Autoformer MAE |
|---|---|---|---|---|---|---|---|---|---|---|---|---|---|---|---|---|---|---|---|---|---|---|---|
| ETTh1 | 96 | **0.364** | **0.392** | **0.364** | 0.397 | 0.385 | 0.412 | 0.379 | 0.415 | 0.387 | 0.418 | 0.370 | 0.400 | 0.376 | 0.397 | 0.452 | 0.463 | 0.406 | 0.420 | 0.376 | 0.419 | 0.467 | 0.499 |
| | 192 | **0.396** | **0.412** | 0.405 | 0.424 | 0.424 | 0.438 | 0.415 | 0.438 | 0.416 | 0.437 | 0.413 | 0.429 | 0.416 | 0.418 | 0.474 | 0.477 | 0.457 | 0.450 | 0.420 | 0.448 | 0.492 | 0.523 |
| | 336 | **0.417** | **0.426** | 0.427 | 0.439 | 0.460 | 0.463 | 0.440 | 0.454 | 0.434 | 0.450 | 0.422 | 0.440 | 0.442 | 0.433 | 0.493 | 0.489 | 0.457 | 0.459 | 0.459 | 0.465 | 0.519 | 0.531 |
| | 720 | **0.431** | **0.452** | 0.439 | 0.459 | 0.486 | 0.487 | 0.482 | 0.482 | 0.447 | 0.473 | 0.447 | 0.468 | 0.477 | 0.456 | 0.560 | 0.534 | 0.493 | 0.498 | 0.506 | 0.507 | 0.589 | 0.560 |
| | Avg. | **0.402** | **0.420** | 0.409 | 0.430 | 0.439 | 0.450 | 0.429 | 0.447 | 0.421 | 0.444 | 0.413 | 0.434 | 0.428 | 0.426 | 0.495 | 0.491 | 0.453 | 0.457 | 0.440 | 0.460 | 0.517 | 0.528 |
| ETTh2 | 96 | **0.272** | **0.335** | 0.277 | 0.343 | 0.293 | 0.352 | 0.343 | 0.398 | 0.297 | 0.349 | 0.274 | 0.337 | 0.285 | 0.342 | 0.340 | 0.374 | 0.315 | 0.373 | 0.358 | 0.397 | 0.358 | 0.397 |
| | 192 | **0.340** | **0.378** | 0.348 | 0.391 | 0.359 | 0.395 | 0.376 | 0.420 | 0.380 | 0.400 | 0.341 | 0.382 | 0.354 | 0.389 | 0.402 | 0.414 | 0.376 | 0.407 | 0.429 | 0.439 | 0.435 | 0.451 |
| | 336 | 0.369 | 0.405 | 0.375 | 0.418 | 0.392 | 0.423 | 0.399 | 0.435 | 0.428 | 0.432 | **0.329** | **0.384** | 0.373 | 0.407 | 0.452 | 0.452 | 0.376 | 0.419 | 0.496 | 0.487 | 0.454 | 0.475 |
| | 720 | 0.404 | 0.436 | 0.409 | 0.458 | 0.425 | 0.451 | 0.419 | 0.457 | 0.427 | 0.445 | **0.379** | **0.422** | 0.406 | 0.441 | 0.462 | 0.468 | 0.404 | 0.438 | 0.463 | 0.474 | 0.432 | 0.454 |
| | Avg. | 0.346 | 0.388 | 0.352 | 0.402 | 0.367 | 0.405 | 0.384 | 0.428 | 0.383 | 0.406 | **0.331** | **0.381** | 0.355 | 0.395 | 0.414 | 0.427 | 0.368 | 0.409 | 0.436 | 0.449 | 0.432 | 0.454 |
| ETTm1 | 96 | 0.292 | 0.339 | 0.290 | 0.341 | 0.301 | 0.357 | **0.289** | 0.348 | 0.311 | 0.365 | 0.293 | 0.346 | 0.292 | 0.346 | 0.338 | 0.375 | 0.333 | 0.382 | 0.379 | 0.419 | 0.466 | 0.466 |
| | 192 | **0.332** | **0.364** | 0.337 | 0.369 | 0.341 | 0.377 | **0.332** | 0.375 | 0.353 | 0.390 | 0.333 | 0.370 | 0.332 | 0.372 | 0.371 | 0.387 | 0.372 | 0.401 | 0.426 | 0.441 | 0.504 | 0.496 |
| | 336 | **0.365** | **0.385** | 0.374 | 0.392 | 0.376 | 0.396 | **0.365** | 0.397 | 0.387 | 0.411 | 0.369 | 0.392 | 0.366 | 0.394 | 0.410 | 0.411 | 0.391 | 0.411 | 0.445 | 0.459 | 0.574 | 0.530 |
| | 720 | 0.425 | 0.423 | 0.437 | 0.428 | 0.431 | 0.425 | 0.421 | 0.431 | 0.452 | 0.445 | **0.416** | **0.420** | 0.417 | 0.421 | 0.478 | 0.450 | 0.437 | 0.438 | 0.543 | 0.490 | 0.596 | 0.558 |
| | Avg. | 0.354 | **0.378** | 0.360 | 0.382 | 0.362 | 0.389 | 0.352 | 0.388 | 0.376 | 0.403 | 0.353 | 0.382 | 0.352 | 0.383 | 0.399 | 0.406 | 0.383 | 0.408 | 0.448 | 0.452 | 0.535 | 0.512 |
| ETTm2 | 96 | **0.164** | **0.250** | 0.175 | 0.257 | 0.176 | 0.265 | 0.171 | 0.260 | 0.183 | 0.272 | 0.166 | 0.256 | 0.173 | 0.262 | 0.187 | 0.267 | 0.204 | 0.297 | 0.203 | 0.287 | 0.255 | 0.339 |
| | 192 | **0.221** | 0.289 | 0.242 | 0.301 | 0.231 | 0.305 | 0.228 | **0.230** | 0.250 | 0.315 | 0.223 | 0.296 | 0.229 | 0.301 | 0.249 | 0.309 | 0.250 | 0.323 | 0.269 | 0.328 | 0.279 | 0.335 |
| | 336 | **0.272** | **0.324** | 0.293 | 0.337 | 0.282 | 0.338 | 0.282 | 0.336 | 0.311 | 0.356 | 0.274 | 0.329 | 0.286 | 0.341 | 0.321 | 0.351 | 0.300 | 0.353 | 0.325 | 0.366 | 0.331 | 0.374 |
| | 720 | 0.364 | **0.379** | 0.376 | 0.390 | **0.361** | 0.388 | 0.380 | 0.398 | 0.417 | 0.419 | 0.362 | 0.385 | 0.378 | 0.401 | 0.497 | 0.403 | 0.396 | 0.411 | 0.421 | 0.415 | 0.413 | 0.450 |
| | Avg. | **0.255** | 0.310 | 0.271 | 0.321 | 0.262 | 0.324 | 0.265 | **0.306** | 0.290 | 0.340 | 0.256 | 0.316 | 0.266 | 0.326 | 0.314 | 0.332 | 0.288 | 0.346 | 0.304 | 0.349 | 0.320 | 0.374 |
| Electricity | 96 | 0.131 | 0.225 | **0.127** | **0.219** | 0.127 | 0.223 | 0.130 | 0.225 | 0.133 | 0.229 | 0.132 | 0.232 | 0.139 | 0.238 | 0.184 | 0.288 | 0.806 | 0.674 | 0.193 | 0.308 | 0.256 | 0.357 |
| | 192 | 0.147 | 0.240 | 0.145 | **0.236** | 0.144 | 0.239 | 0.150 | 0.244 | 0.158 | 0.258 | 0.151 | 0.250 | 0.153 | 0.251 | 0.192 | 0.295 | 0.816 | 0.682 | 0.201 | 0.315 | 0.291 | 0.376 |
| | 336 | 0.164 | 0.258 | **0.159** | **0.252** | 0.159 | 0.255 | 0.166 | 0.262 | 0.168 | 0.262 | 0.171 | 0.272 | 0.169 | 0.266 | 0.200 | 0.303 | 0.812 | 0.680 | 0.214 | 0.329 | 0.290 | 0.379 |
| | 720 | 0.206 | 0.293 | **0.187** | **0.277** | 0.196 | 0.290 | 0.206 | 0.297 | 0.205 | 0.294 | 0.222 | 0.318 | 0.206 | 0.297 | 0.228 | 0.325 | 0.800 | 0.665 | 0.246 | 0.355 | 0.320 | 0.403 |
| | Avg. | 0.162 | 0.254 | **0.155** | **0.246** | 0.157 | 0.252 | 0.163 | 0.257 | 0.166 | 0.261 | 0.169 | 0.268 | 0.167 | 0.263 | 0.201 | 0.303 | 0.808 | 0.675 | 0.214 | 0.327 | 0.289 | 0.379 |
| Weather | 96 | **0.146** | **0.195** | 0.157 | 0.205 | 0.149 | 0.203 | 0.152 | 0.206 | 0.174 | 0.225 | 0.149 | 0.198 | 0.162 | 0.212 | 0.169 | 0.228 | 0.170 | 0.217 | 0.217 | 0.296 | 0.355 | 0.409 |
| | 192 | **0.189** | **0.237** | 0.206 | 0.250 | 0.192 | 0.244 | 0.198 | 0.249 | 0.227 | 0.268 | 0.194 | 0.241 | 0.204 | 0.248 | 0.222 | 0.269 | 0.213 | 0.254 | 0.276 | 0.336 | 0.421 | 0.450 |
| | 336 | **0.238** | **0.277** | 0.259 | 0.291 | 0.242 | 0.283 | 0.251 | 0.291 | 0.290 | 0.309 | 0.245 | 0.282 | 0.254 | 0.286 | 0.290 | 0.310 | 0.261 | 0.290 | 0.339 | 0.380 | 0.452 | 0.465 |
| | 720 | **0.312** | **0.328** | 0.337 | 0.344 | 0.312 | 0.333 | 0.322 | 0.340 | 0.374 | 0.360 | 0.314 | 0.334 | 0.326 | 0.337 | 0.376 | 0.364 | 0.324 | 0.336 | 0.403 | 0.428 | 0.513 | 0.496 |
| | Avg. | **0.221** | **0.259** | 0.240 | 0.272 | 0.224 | 0.266 | 0.231 | 0.272 | 0.266 | 0.290 | 0.225 | 0.264 | 0.236 | 0.271 | 0.264 | 0.293 | 0.242 | 0.274 | 0.309 | 0.360 | 0.435 | 0.455 |
| | 1st Count | **15** | **16** | 4 | 4 | 4 | 0 | 3 | 1 | 0 | 0 | 3 | 3 | 0 | 0 | 0 | 0 | 0 | 0 | 0 | 0 | 0 | 0 |

# B   Supplementary Experiments and Results

## B.1   Full-Shot Forecasting

**Full-Shot Forecasting.**   For full-shot, we evaluate against Timer-XL, UniTST, CycleNet, iTransformer, PatchTST, GPT4TS, TimesNet, and Autoformer (Liu et al., 2024c;a; Lin et al., 2024a; Liu et al., 2023; Nie et al., 2023; Zhou et al., 2023; Wu et al., 2022; 2021). Comparisons to additional MoE-based models are given in App. B.1. We evaluate Super-Linear against leading in-domain TSF models, with detailed results provided in Tab. 4. Results for CycleNet, GPT4TS, and PatchTST are drawn from their original papers, while others follow the evaluation setup of Liu et al. (2024c). All baselines are assessed using comparable lookback windows: $L = 512$ for PatchTST and GPT4TS, $L = 720$ for CycleNet, and $L = 672$ for the remaining models. Super-Linear uses a fixed lookback of $L = 512$ and selects the best-performing model based on validation MSE across top-$k$ experts, where $k \in 6, 8, 10, 12, 20$, with all other hyperparameters unchanged. It achieves the best MSE and MAE in **15** and **16** benchmarks, respectively—outperforming all competitors. On average, Super-Linear yields a **2.5%** and **3.9%** reduction in MSE compared to Timer-XL and CycleNet, respectively. Notably, it requires only a single training per dataset, with longer horizons handled autoregressively. Further, Stage 2 is repeated across three random seeds while training only on the target dataset and not on the pre-trained data.

### B.2 MoE and Linear Model Full-Shot Comparison

In this section, we compare the full-shot forecasting results presented in Table 4 with both traditional linear models—DLinear and MTLinear (Zeng et al., 2023; Nochumsohn et al., 202t)—and Mixture-of-Experts (MoE)-based models, including FreqMoE and MoLE (Liu, 2025; Ni et al., 2024). These comparisons highlight the performance gains introduced by our proposed model, Super-Linear, in a full-data setting. As shown in Table 5, Super-Linear consistently outperforms the baselines, achieving the lowest Mean Squared Error (MSE) on 19 out of 24 benchmarks and the lowest Mean Absolute Error (MAE) on 17 out of 24. These results underscore the effectiveness of combining a mixture of pre-trained frequency experts and non-pretrained trainable experts for time series forecasting.

Table 5: Super-Linear full-shot comparison to Linear and MoE based models. All models utilize a lookback of 512.

| Dataset | Horizon | Super-Linear MSE | Super-Linear MAE | FreqMoE MSE | FreqMoE MAE | MTLinear MSE | MTLinear MAE | MoLE MSE | MoLE MAE | DLinear MSE | DLinear MAE |
|---|---|---|---|---|---|---|---|---|---|---|---|
| ETTh1 | 96 | **0.364** | **0.392** | 0.690 | 0.568 | 0.411 | 0.430 | 0.379 | 0.406 | 0.371 | 0.399 |
| | 192 | **0.396** | **0.412** | 0.713 | 0.584 | 0.444 | 0.453 | 0.410 | 0.425 | 0.409 | 0.425 |
| | 336 | **0.417** | **0.426** | 0.717 | 0.595 | 0.458 | 0.458 | 0.432 | 0.439 | 0.438 | 0.444 |
| | 720 | **0.431** | **0.452** | 0.788 | 0.650 | 0.475 | 0.484 | 0.464 | 0.474 | 0.480 | 0.497 |
| | Avg. | **0.402** | **0.420** | 0.727 | 0.599 | 0.447 | 0.456 | 0.421 | 0.436 | 0.424 | 0.441 |
| ETTh2 | 96 | **0.272** | **0.335** | 0.376 | 0.426 | 0.298 | 0.360 | 0.274 | 0.341 | 0.300 | 0.366 |
| | 192 | 0.340 | 0.378 | 0.401 | 0.441 | 0.358 | 0.399 | **0.325** | **0.377** | 0.386 | 0.419 |
| | 336 | 0.369 | 0.405 | 0.412 | 0.449 | 0.380 | 0.418 | **0.353** | **0.401** | 0.490 | 0.486 |
| | 720 | 0.404 | 0.436 | 0.469 | 0.484 | 0.401 | 0.438 | **0.393** | **0.431** | 0.784 | 0.629 |
| | Avg. | 0.346 | **0.388** | 0.414 | 0.450 | 0.359 | 0.404 | **0.336** | 0.388 | 0.490 | 0.475 |
| ETTm1 | 96 | **0.292** | **0.339** | 0.500 | 0.462 | 0.310 | 0.354 | 0.306 | 0.352 | 0.308 | 0.351 |
| | 192 | **0.332** | **0.364** | 0.655 | 0.541 | 0.350 | 0.377 | 0.347 | 0.374 | 0.340 | 0.370 |
| | 336 | **0.365** | **0.385** | 0.652 | 0.542 | 0.382 | 0.395 | 0.377 | 0.391 | 0.373 | 0.393 |
| | 720 | **0.425** | 0.423 | 0.654 | 0.549 | 0.440 | 0.427 | 0.435 | 0.422 | 0.428 | 0.425 |
| | Avg. | **0.354** | **0.378** | 0.615 | 0.524 | 0.370 | 0.388 | 0.366 | 0.385 | 0.362 | 0.385 |
| ETTm2 | 96 | **0.164** | **0.250** | 0.272 | 0.349 | 0.172 | 0.261 | 0.169 | 0.257 | 0.179 | 0.274 |
| | 192 | **0.221** | **0.289** | 0.314 | 0.372 | 0.227 | 0.298 | 0.227 | 0.297 | 0.253 | 0.332 |
| | 336 | **0.272** | **0.324** | 0.348 | 0.389 | 0.284 | 0.335 | 0.299 | 0.340 | 0.309 | 0.369 |
| | 720 | **0.364** | **0.379** | 0.439 | 0.442 | 0.373 | 0.392 | 0.380 | 0.393 | 0.398 | 0.419 |
| | Avg. | **0.255** | **0.310** | 0.343 | 0.388 | 0.264 | 0.322 | 0.269 | 0.322 | 0.285 | 0.349 |
| Electricity | 96 | **0.131** | **0.225** | 0.537 | 0.543 | 0.140 | 0.238 | 0.144 | 0.241 | 0.140 | 0.239 |
| | 192 | **0.147** | **0.240** | 0.532 | 0.538 | 0.169 | 0.272 | 0.159 | 0.257 | 0.156 | 0.257 |
| | 336 | **0.164** | **0.258** | 0.573 | 0.566 | 0.186 | 0.287 | 0.182 | 0.276 | 0.169 | 0.271 |
| | 720 | 0.206 | 0.293 | 0.520 | 0.529 | 0.224 | 0.318 | **0.200** | **0.287** | 0.206 | 0.306 |
| | Avg. | **0.162** | **0.254** | 0.540 | 0.544 | 0.180 | 0.279 | 0.171 | 0.265 | 0.168 | 0.268 |
| Weather | 96 | **0.146** | **0.195** | 0.191 | 0.244 | 0.148 | 0.201 | 0.221 | 0.218 | 0.173 | 0.232 |
| | 192 | 0.189 | 0.237 | 0.262 | 0.306 | 0.195 | 0.244 | **0.188** | **0.236** | 0.216 | 0.272 |
| | 336 | **0.238** | **0.277** | 0.299 | 0.331 | 0.245 | 0.282 | 0.240 | 0.276 | 0.259 | 0.307 |
| | 720 | **0.312** | **0.328** | 0.366 | 0.380 | 0.315 | 0.333 | 0.313 | 0.330 | 0.321 | 0.359 |
| | Avg. | **0.221** | **0.259** | 0.279 | 0.315 | 0.226 | 0.265 | 0.240 | 0.265 | 0.242 | 0.292 |
| | 1st Count. | **19** | **17** | 0 | 0 | 0 | 0 | 5 | 6 | 0 | 0 |

**Setup.** All evaluations are averaged over three runs with different random seeds and use a lookback window of 512 time steps. For fairness, we reimplement and reproduce results for FreqMoE, MTLinear, MoLE, and DLinear using the same lookback length. A grid search is conducted over learning rates 0.01, 0.005, 0.001, 0.05, selecting the best configuration based on validation loss. For MTLinear, we adopt the learning rate of 0.01 as recommended by the original authors.

Table 6: Super-Linear zero-shot comparison to TTM. All models utilize a lookback of 512.

| Dataset | Horizon | Super-Linear | TTM | TTM-R2 |
|---|---|---|---|---|
| ETTm1 | 96 | **0.317** | 0.413 | 0.415 |
| ETTm1 | 192 | **0.357** | 0.476 | 0.380 |
| ETTm1 | 336 | **0.393** | 0.553 | 0.402 |
| ETTm1 | 720 | 0.459 | 0.737 | **0.446** |
| ETTm1 | Avg. | **0.382** | 0.545 | 0.411 |
| ETTm2 | 96 | **0.179** | 0.187 | 0.186 |
| ETTm2 | 192 | **0.240** | 0.261 | 0.246 |
| ETTm2 | 336 | **0.293** | 0.323 | 0.324 |
| ETTm2 | 720 | 0.410 | 0.436 | **0.406** |
| ETTm2 | Avg. | **0.280** | 0.302 | 0.290 |
| ETTh2 | 96 | **0.279** | 0.285 | 0.286 |
| ETTh2 | 192 | **0.333** | 0.341 | 0.346 |
| ETTh2 | 336 | **0.354** | 0.383 | 0.385 |
| ETTh2 | 720 | **0.382** | 0.441 | 0.419 |
| ETTh2 | Avg. | **0.337** | 0.362 | 0.359 |
| ETTh1 | 96 | 0.369 | 0.365 | **0.363** |
| ETTh1 | 192 | 0.399 | 0.393 | **0.387** |
| ETTh1 | 336 | 0.416 | 0.415 | **0.402** |
| ETTh1 | 720 | **0.424** | 0.538 | 0.475 |
| ETTh1 | Avg. | **0.402** | 0.428 | 0.407 |
| Electricity | 96 | **0.141** | 0.169 | 0.170 |
| Electricity | 192 | **0.157** | 0.196 | 0.194 |
| Electricity | 336 | **0.175** | 0.209 | 0.213 |
| Electricity | 720 | **0.217** | 0.264 | 0.260 |
| Electricity | Avg. | **0.172** | 0.209 | 0.209 |
| Weather | 96 | 0.159 | 0.154 | **0.152** |
| Weather | 192 | 0.207 | 0.203 | **0.195** |
| Weather | 336 | 0.261 | **0.256** | **0.256** |
| Weather | 720 | 0.333 | 0.329 | **0.319** |
| Weather | Avg. | 0.240 | 0.235 | **0.230** |
| Traffic | 96 | **0.414** | 0.518 | 0.509 |
| Traffic | 192 | **0.430** | 0.548 | 0.538 |
| Traffic | 336 | **0.449** | 0.550 | 0.571 |
| Traffic | 720 | **0.551** | 0.605 | 0.617 |
| Traffic | Avg. | **0.461** | 0.555 | 0.559 |

### B.3 Super-Linear comprehensive ZS Comparison to TTM

**LTSF** : In this section, we present a detailed comparison with the TTM model. Since the original work does not report MAE results, we provide an MSE-only comparison. We evaluate Super-Linear against both the TTM-Q results reported in the original paper and TTM-R2, which is available on Hugging Face. Across seven datasets, Super-Linear achieves superior performance in six cases on average as presented in Table 6.

**GIFT-Eval** : As shown in Figure 5, Super-Linear achieves a substantial performance gain of up to **16%** over TTM-R2, with a MASE score of **0.857** compared to **1.020**. This highlights the robustness of Super-Linear across diverse domains and sampling rates.

## B.4 Comparison to LLM based models

In Tab.7 We compare Super-Linear (full-shot) to recent LLM- and VLM-based forecasting models, namely Time-LLM (Jin et al., 2023), Time-VLM (Zhong et al., 2025), and TimeCMA (Liu et al., 2025a). Super-Linear achieves the best performance in 8 configurations in terms of MSE and 10 configurations in terms of MAE, which is competitive with Time-LLM (10 best MSE scores and 8 best MAE scores). Notably, this performance is obtained with a substantially smaller model: Super-Linear contains only 2.5M parameters, compared to 143M parameters for Time-VLM and 3,405M parameters for Time-LLM. Furthermore, Super-Linear is trained only once on the 96-step forecasting horizon, and the longer horizons (192, 336, and 720) are obtained via inference using the same trained model, without retraining. In contrast, the competing methods train separate models for each dataset and horizon configuration. Additionally, in our work we strictly follow the standard machine learning protocol of selecting hyperparameters based on validation-set performance. These prior LLM- and VLM-based forecasting works report different hyperparameter configurations across datasets and horizons, but do not describe a systematic validation-driven search procedure for selecting these configurations, providing instead only hyperparameter sensitivity analyses. Overall, Super-Linear achieves competitive performance while being orders of magnitude smaller, requiring substantially less training, and adhering to a consistent model selection protocol.

Table 7: Super-Linear full-shot comparison to to LLM-based forecasting models.

| Dataset | Horizon | Super-Linear MSE | Super-Linear MAE | TimeCMA MSE | TimeCMA MAE | Time-VLM MSE | Time-VLM MAE | Time-LLM MSE | Time-LLM MAE |
|---|---|---|---|---|---|---|---|---|---|
| ETTh1 | 96 | 0.364 | 0.392 | 0.373 | 0.391 | **0.361** | **0.386** | 0.362 | 0.392 |
| | 192 | **0.396** | **0.412** | 0.427 | 0.421 | 0.397 | 0.415 | 0.398 | 0.418 |
| | 336 | **0.417** | 0.426 | 0.458 | 0.448 | 0.420 | **0.421** | 0.430 | 0.427 |
| | 720 | **0.431** | **0.452** | 0.449 | 0.460 | 0.441 | 0.458 | 0.442 | 0.457 |
| | Avg. | **0.402** | **0.420** | 0.427 | 0.430 | 0.405 | **0.420** | 0.408 | 0.424 |
| ETTh2 | 96 | 0.272 | 0.335 | 0.286 | 0.336 | **0.267** | 0.335 | 0.268 | **0.328** |
| | 192 | 0.340 | 0.378 | 0.363 | 0.387 | **0.326** | **0.373** | 0.329 | 0.375 |
| | 336 | 0.369 | **0.405** | 0.406 | 0.421 | **0.357** | 0.406 | 0.368 | 0.409 |
| | 720 | 0.404 | 0.436 | 0.417 | 0.438 | 0.412 | 0.449 | **0.372** | **0.420** |
| | Avg. | 0.346 | 0.388 | 0.368 | 0.396 | 0.340 | 0.391 | **0.334** | **0.383** |
| ETTm1 | 96 | 0.292 | 0.339 | 0.312 | 0.351 | 0.304 | 0.346 | **0.272** | **0.334** |
| | 192 | 0.332 | 0.364 | 0.361 | 0.378 | 0.332 | 0.366 | **0.310** | **0.358** |
| | 336 | 0.365 | 0.385 | 0.392 | 0.401 | 0.364 | **0.383** | **0.352** | 0.384 |
| | 720 | 0.425 | 0.423 | 0.453 | 0.438 | 0.402 | **0.410** | **0.383** | 0.411 |
| | Avg. | 0.354 | 0.378 | 0.380 | 0.392 | 0.351 | 0.376 | **0.329** | **0.372** |
| ETTm2 | 96 | 0.164 | **0.250** | 0.173 | 0.258 | **0.160** | **0.250** | 0.161 | 0.253 |
| | 192 | 0.221 | **0.289** | 0.238 | 0.301 | **0.215** | 0.291 | 0.219 | 0.293 |
| | 336 | 0.272 | **0.324** | 0.297 | 0.338 | **0.270** | 0.325 | 0.271 | 0.329 |
| | 720 | 0.364 | 0.379 | 0.393 | 0.394 | **0.348** | **0.378** | 0.352 | 0.379 |
| | Avg. | 0.255 | **0.310** | 0.275 | 0.323 | **0.248** | 0.311 | 0.251 | 0.314 |
| Electricity | 96 | **0.131** | 0.225 | 0.143 | 0.238 | 0.142 | 0.245 | **0.131** | **0.224** |
| | 192 | **0.147** | **0.240** | 0.161 | 0.259 | 0.157 | 0.260 | 0.152 | 0.241 |
| | 336 | 0.164 | 0.258 | 0.169 | 0.261 | 0.174 | 0.276 | **0.160** | **0.248** |
| | 720 | 0.206 | **0.293** | 0.219 | 0.315 | 0.214 | 0.308 | **0.192** | 0.298 |
| | Avg. | 0.162 | 0.254 | 0.173 | 0.268 | 0.172 | 0.272 | **0.159** | **0.253** |
| Weather | 96 | **0.146** | **0.195** | 0.167 | 0.211 | 0.148 | 0.200 | 0.147 | 0.201 |
| | 192 | **0.189** | 0.237 | 0.212 | 0.253 | 0.193 | 0.240 | **0.189** | **0.234** |
| | 336 | **0.238** | **0.277** | 0.270 | 0.292 | 0.243 | 0.281 | 0.262 | 0.279 |
| | 720 | 0.312 | 0.328 | 0.350 | 0.348 | 0.312 | 0.332 | **0.304** | **0.316** |
| | Avg. | **0.221** | 0.259 | 0.250 | 0.276 | 0.224 | 0.263 | 0.225 | **0.258** |
| | 1st Count | 8 | **10** | 0 | 0 | 8 | 7 | **10** | 8 |

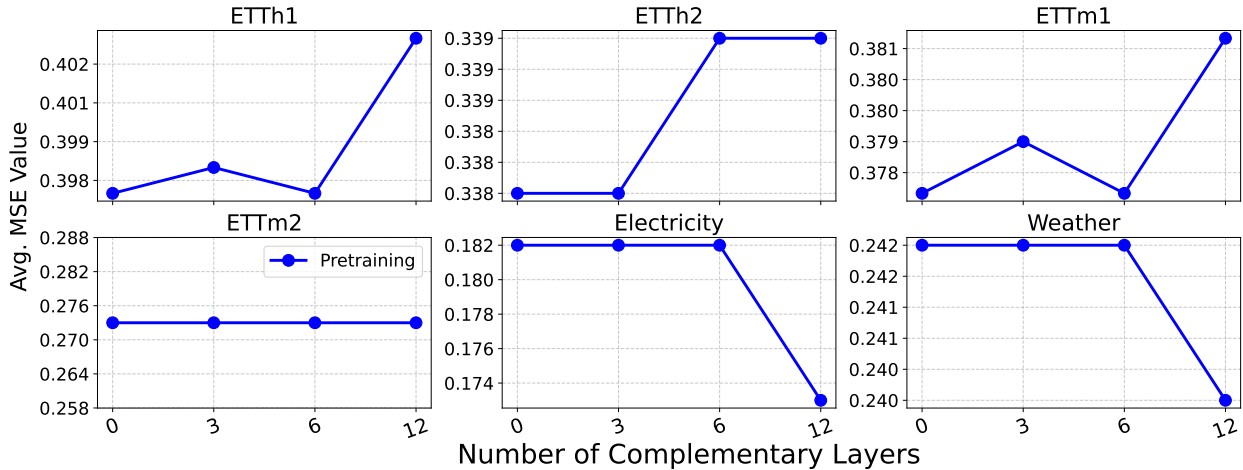

Figure 8: Complementary layer ablation for zero-shot evaluations across various datasets. The value 0 reflects Super-Linear with no complementary layers, whereas 12 represents Super-Linear with 12 complementary layers.

Table 8: Comparison of different frequency selection strategies. Random: Select random frequencies. Uniform: select equally spaced frequencies, and ours: Select natural and common frequencies as discussed in D.3.

|             | Ours      | Random  | Uniform |
|-------------|-----------|---------|---------|
| ETTm1       | **0.317** | 0.352   | 0.369   |
| ETTm2       | **0.179** | 0.188   | 0.194   |
| ETTh2       | **0.279** | 0.283   | 0.301   |
| ETTh1       | **0.369** | 0.366   | 0.417   |
| Traffic     | **0.414** | 0.424   | 0.477   |
| Weather     | **0.159** | 0.165   | 0.161   |
| Electricity | **0.141** | 0.152   | 0.182   |

## B.5 Complementary Experts Ablation

In this section, we evaluate the impact of the number of complementary layers used during pre-training across various datasets. The original pre-trained model, whose results are shown in Table 1, employs $N_c = 12$ complementary layers. Here, we explore a range of values for $N_c$ to assess how this design choice affects performance. The results reveal that there is no universally optimal configuration: for datasets such as ETTh1, ETTh2, and ETTm1, introducing additional complementary layers can even degrade performance. In contrast, for datasets like Electricity and Weather, adding these layers yields significant improvements. This suggests that the effectiveness of complementary layers is dataset-dependent. One limitation of the current approach is the exclusive use of linear modules as complementary components—introducing non-linear transformations may offer further improvements and is a path for future exploration.

**Experimental Setup.** We follow the same training protocol described in Figure 4, and report average performance across four forecasting horizons: 96, 192, 336, and 720.

## B.6 Frequency Choice

In this section, we compare three frequency selection strategies: uniformly spaced frequencies, randomly selected frequencies, and our method. The results are given in Table 8. Both the uniform and random baselines select 37 frequencies from the range $[\frac{1}{512}, \frac{1}{4}]$. We chose this interval because it contains many

common natural frequencies (e.g., hourly, daily, monthly), giving these baselines a slight advantage. The rest of the training pipeline remains unchanged to ensure a fair comparison.

The results clearly show that selecting natural, commonly occurring frequencies, as done in our approach, yields substantially more consistent performance. This aligns with the theoretical *bias–complexity tradeoff* discussed in Section 4.3. The large errors observed in the uniform and random settings stem from *frequency approximation error*, which increases when the signal's energy lies outside an expert's frequency range.

### B.7 Full-Shot Results With Standard Deviation

Table 9 reports the standard deviation of the results presented in Table 4, calculated over three runs with different random seeds.

Table 9: Super-Linear full-shot with the standard deviation.

| Dataset | Horizon | Super-Linear MSE | Super-Linear MAE |
|---|---|---|---|
| ETTh1 | 96 | $0.364 \pm 0.001$ | $0.392 \pm 0.001$ |
| ETTh1 | 192 | $0.396 \pm 0.001$ | $0.412 \pm 0.001$ |
| ETTh1 | 336 | $0.417 \pm 0.002$ | $0.426 \pm 0.001$ |
| ETTh1 | 720 | $0.431 \pm 0.002$ | $0.452 \pm 0.002$ |
| ETTh2 | 96 | $0.272 \pm 0.001$ | $0.335 \pm 0.003$ |
| ETTh2 | 192 | $0.340 \pm 0.005$ | $0.378 \pm 0.004$ |
| ETTh2 | 336 | $0.369 \pm 0.010$ | $0.405 \pm 0.005$ |
| ETTh2 | 720 | $0.404 \pm 0.013$ | $0.436 \pm 0.011$ |
| ETTm1 | 96 | $0.292 \pm 0.002$ | $0.339 \pm 0.001$ |
| ETTm1 | 192 | $0.332 \pm 0.001$ | $0.364 \pm 0.001$ |
| ETTm1 | 336 | $0.365 \pm 0.001$ | $0.385 \pm 0.002$ |
| ETTm1 | 720 | $0.425 \pm 0.001$ | $0.423 \pm 0.003$ |
| ETTm2 | 96 | $0.164 \pm 0.001$ | $0.250 \pm 0.001$ |
| ETTm2 | 192 | $0.221 \pm 0.002$ | $0.289 \pm 0.001$ |
| ETTm2 | 336 | $0.272 \pm 0.003$ | $0.324 \pm 0.001$ |
| ETTm2 | 720 | $0.364 \pm 0.002$ | $0.379 \pm 0.001$ |
| Electricity | 96 | $0.131 \pm 0.000$ | $0.225 \pm 0.001$ |
| Electricity | 192 | $0.147 \pm 0.001$ | $0.240 \pm 0.000$ |
| Electricity | 336 | $0.164 \pm 0.000$ | $0.258 \pm 0.001$ |
| Electricity | 720 | $0.206 \pm 0.001$ | $0.293 \pm 0.001$ |
| Weather | 96 | $0.146 \pm 0.001$ | $0.195 \pm 0.001$ |
| Weather | 192 | $0.189 \pm 0.000$ | $0.237 \pm 0.001$ |
| Weather | 336 | $0.238 \pm 0.001$ | $0.277 \pm 0.001$ |
| Weather | 720 | $0.312 \pm 0.001$ | $0.328 \pm 0.001$ |

### B.8 Full Ablation Results

In this section the full results of the LTSF ablation elements from Table 2.

Table 10: Super-Linear full-shot comparison to Linear and MoE based models. All models utilize a lookback of 512 and a forecast horizon of 96.

| Dataset | full model MSE | full model MAE | WO Compl MSE | WO Compl MAE | WO Mean Naive MSE | WO Mean Naive MAE | One Stage MSE | One Stage MAE | WO Tr Resamp MSE | WO Tr Resamp MAE |
|---|---|---|---|---|---|---|---|---|---|---|
| ETTh1 | 0.369 | 0.392 | 0.365 | 0.393 | 0.365 | 0.392 | 0.385 | 0.410 | 0.401 | 0.420 |
| ETTh2 | 0.279 | 0.342 | 0.279 | 0.342 | 0.278 | 0.342 | 0.284 | 0.350 | 0.296 | 0.359 |
| ETTm1 | 0.317 | 0.352 | 0.314 | 0.350 | 0.315 | 0.351 | 0.352 | 0.376 | 1.117 | 0.622 |
| ETTm2 | 0.179 | 0.265 | 0.179 | 0.265 | 0.178 | 0.264 | 0.189 | 0.274 | 0.232 | 0.314 |
| Electricity | 0.141 | 0.239 | 0.151 | 0.249 | 0.150 | 0.248 | 0.154 | 0.255 | 0.209 | 0.297 |
| Traffic | 0.414 | 0.296 | 0.425 | 0.317 | 0.425 | 0.317 | 0.447 | 0.323 | 0.529 | 0.354 |
| Weather | 0.159 | 0.211 | 0.160 | 0.212 | 0.162 | 0.214 | 0.164 | 0.215 | 0.175 | 0.226 |

### B.9 Varying Sampling Rates

Extending the discussion from Sec. 4.2 on zero-shot generalization across varying sampling rates, we include a comparison on the GIFT-Eval benchmark using the BizItObs dataset at a 10-second sampling rate. This

Table 11: We compare the performance of Super-Linear on datasets with sampling frequencies outside its training regime, specifically at the 10-second sampling rate on the BizItObs dataset from GIFT-Eval.

| dataset | TiRex | Sundial | Super-Linear | Timer | Chronos Base | TTM R2 | Chronos Bolt Small | Moirai Base | TimesFM |
|---|---|---|---|---|---|---|---|---|---|
| bizitobs application 10S long | 3.40 | 3.71 | 3.81 | 8.51 | 9.25 | 9.59 | 9.65 | 13.50 | 16.30 |
| bizitobs application 10S medium | 2.64 | 2.86 | 3.06 | 7.47 | 9.87 | 9.02 | 9.15 | 12.80 | 11.20 |
| bizitobs application 10S short | 1.21 | 1.43 | 1.55 | 3.75 | 3.01 | 4.21 | 5.41 | 5.32 | 4.36 |
| bizitobs service 10S long | 1.49 | 1.46 | 1.57 | 4.25 | 4.22 | 5.34 | 4.85 | 6.08 | 7.77 |
| bizitobs service 10S medium | 1.17 | 1.27 | 1.39 | 4.26 | 4.58 | 5.12 | 4.64 | 5.99 | 6.46 |
| bizitobs service 10S short | 0.82 | 0.84 | 0.94 | 2.17 | 1.88 | 2.73 | 2.90 | 3.43 | 2.90 |
| Average | 1.79 | 1.93 | 2.05 | 5.07 | 5.47 | 6.00 | 6.10 | 7.85 | 8.16 |

sampling rate is rarely present in the training corpora of the evaluated foundation models, and for Super-Linear, it also falls outside the range of its frequency experts. Thus, this dataset serves as an additional benchmark for evaluating generalization to unseen frequencies. The results show that Super-Linear achieves an average score of 2.05, slightly lower than TiREX and Sundial, but substantially better than Chronos-Bolt ($\sim$3) and Timer ($\sim$2.5), highlighting the strong generalization capability of Super-Linear on uncommon sampling rates.

## B.10 Varying Input Lookbacks

$\mathbf{L_{input}} < \mathbf{512}$. Although Super-Linear is a linear forecaster, it includes a simple built-in heuristic to handle variable lookback lengths. When the input lookback $L_{input}$ is shorter than the training length ($L_{train} = 512$), the series is upsampled via linear interpolation to match 512. After inference, the output is downsampled back to the original resolution. This process implicitly modifies the signal's frequency, but Super-Linear remains effective due to its robustness to frequency variations. The resampling factor is computed as:

$$\text{resample factor} = \left\lceil \frac{L_{train}}{L_{input}} \right\rceil$$

To revert the output, we interpolate with a factor of $\frac{1}{\text{resample factor}}$. This approach enables Super-Linear to effectively handle a range of input lengths. Examples below illustrate forecasting with lookbacks of 384, 256, and 128 on the Electricity and Weather datasets.

$\mathbf{L_{input}} > \mathbf{512}$. The long lookback resample search algorithm adaptively downsamples time series by evaluating a set of candidate scales (default: $[2, 4, 6]$). For each sequence, it computes the power spectral density (PSD) to estimate signal energy, uses the expert distribution of Super-Linear to assess selection uncertainty, and applies linear interpolation for resampling. The objective is to minimize both energy loss and distributional entropy. To this end, we define a scoring function $score = entropy + \lambda \cdot penalty$, and select the resampling scale with the lowest score. A detailed description of the algorithm is provided in 1. Here is an overview of the flow of the algorithm:

- **Lines 1-3**, Compute the noramlized periodogram $I_{norm}$, and the maximal scale factor $s_{max}$ allowed such that the energy loss will not surpass $E_{max\ loss}$.

- **Lines 4-10**, create a dictionary $A_{penalty}$ that saves the estimated energy loss incured by each scale parameter $s_i \in S$.

- **Lines 13-23**, Search for the best sacle parameter that ensures minimal enrtopy score energy loss with the term $score = entropy(G(X_{interp}))\ + \lambda A_{penalty}[s]$.

- **Line 24** Finally, return the interpolated input $X_{output}$ and the corresponding scale factor $s_{output}$, which will be used for resampling the output from Super-Linear to the original granularity.

In summary, the algorithm handles long-context inputs containing low-frequency components that may fall outside the range of Super-Linear's frequency experts. It selects an appropriate resampling factor that shifts the input frequencies into a space the model can represent, while preserving the signal's energy to avoid information loss.

---

**Algorithm 1** Long Lookback Resample Search Algorithm

---

**Require:** Input $X \in \mathbb{R}^L$, scales $S = \{s_1, s_2, \ldots\}$, energy loss penalty scaler $\lambda$, a pre-trained Super-Linear model, and a $E_{max\ loss}$ parameter to act as a hard maximal energy loss constrain.

1: $I_{norm} \leftarrow \frac{I(X - \bar{X})}{\|I(X - \bar{X})\|_1} \in \mathbb{R}^M$, Periodogram normalization.

2: $j^* \leftarrow max\{j | E_j \leq 1 - E_{max\ loss}\}$, where $E_j = \sum_1^j I_{norm}^j$, and $E_M = 1$ which denotes the total normalized energy.

3: $s_{max} \leftarrow \lfloor \frac{\omega_{j^*}}{\omega_{min}} \rfloor$, for the maximal scale factor, $\omega_{min}$ marks the highest frequency experts.

4: $A_{penalty} \leftarrow \{\ \}$, a dictionary to track the potential energy loss of each scale $s$.

5: **for** $s \in S, s > 1$ **do**

6:      $\omega_{\text{Nyq}} \leftarrow \frac{0.5}{s}$, marks the threshold from which higher frequencies will be lost.

7:      $I_s \leftarrow \{I_{norm}^j \mid \omega_j > \omega_{\text{Nyq}}\} \in \mathbb{R}^{<M}$,

8:      $E_{\text{lost}} = \|I_s\|_1$,

9:      $A_{penalty}[s] \leftarrow \frac{E_{\text{lost}}}{E_{\text{total}} + \varepsilon}, \quad \varepsilon \approx 10^{-10}$, the fraction of normalized energy loss to be used as a penalty.

10: **end for**

11: $X_{output} \leftarrow X$

12: $s_{output} \leftarrow 1$

13: $score_{best} \leftarrow entropy(G(X))$

14: **for** $s \in S, s > 1$ **do**

15:      $X_{interp} \leftarrow interpolate(X, \frac{1}{s}) \in \mathbb{R}^{L_{interp}}$

16:      **if** $L_{train} \leq L_{interp}$ and $s \leq s_{max}$ **then**

17:          $s \leftarrow entropy(G(X_{interp})) + \lambda A_{penalty}[s]$

18:          **if** $score < score_{best}$ **then**

19:              $s_{output} \leftarrow s$

20:              $X_{output} \leftarrow X_{interp}$

21:          **end if**

22:      **end if**

23: **end for**

24: **return** $X_{output} \in \mathbb{R}^{L_{interp}}, s_{output}$

---

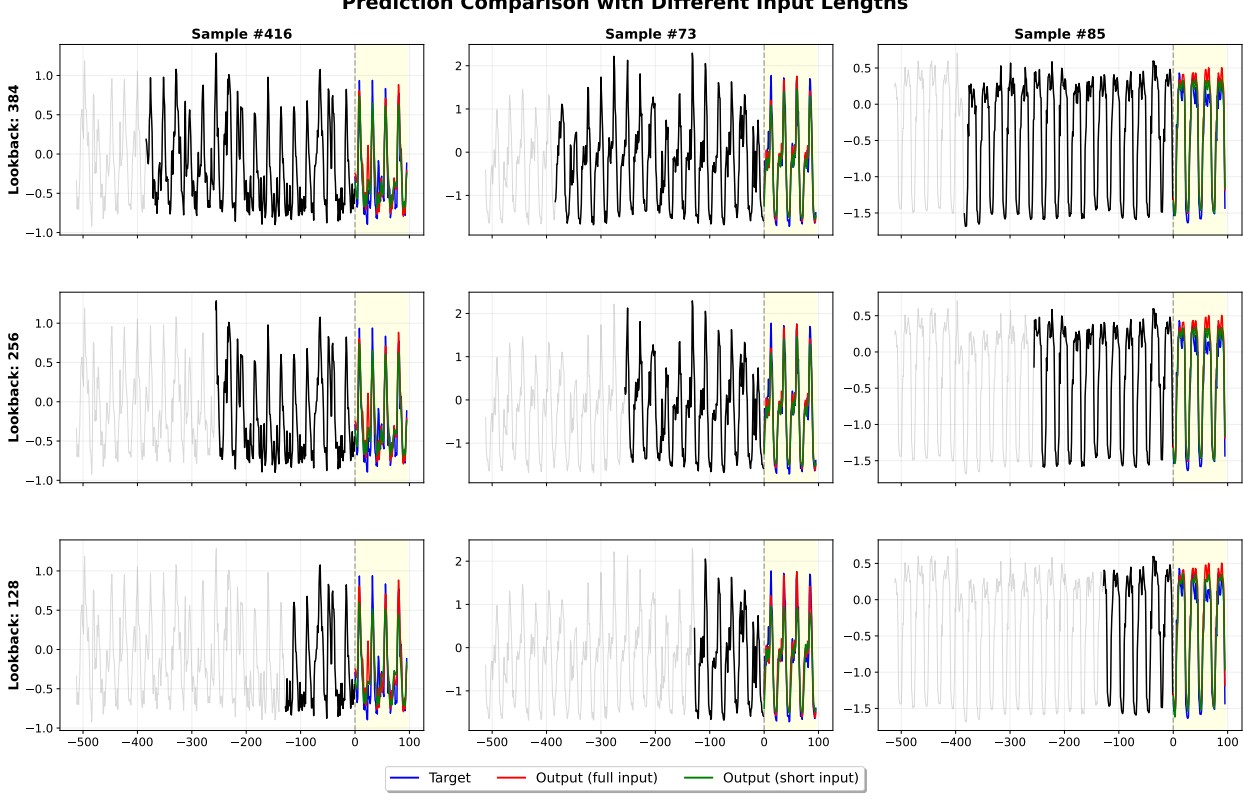

Figure 9: Super-Linear forecasting illustration with varying input lookbacks for the Electricity dataset.

## B.11 Super-Linear Flexibility: Trade-offs and Limitations

A defining property of TSFMs is flexibility across varying look-back lengths and forecast horizons. Super-Linear achieves this via (i) resampling-based alignment of the input window to the fixed training length $L_{\text{train}} = 512$, and (ii) autoregressive roll-out for long horizons. While effective, these mechanisms introduce controlled but non-negligible trade-offs.

### B.11.1 Varying Look-back Lengths

$L_{\text{input}} < 512$. Short inputs are upsampled via linear interpolation and the forecast is subsequently downsampled back to the original resolution.

**Benefits.** Ensures compatibility with the trained frequency experts while preserving dominant low-frequency structure.

**Limitations.**

- *No new information*: Upsampling increases resolution without adding spectral content.

- *Smoothing bias*: Linear interpolation attenuates high-frequency components.

- *Spectral distortion*: Interpolation may alter energy distribution and increase the residual energy outside the expert span $\mathcal{S}$.

From the bound
$$\|Y(X) - \hat{Y}(X)\| \leq \sqrt{E_\perp(X)} + \gamma \|X\|_2 \|\beta - G\|_1,$$

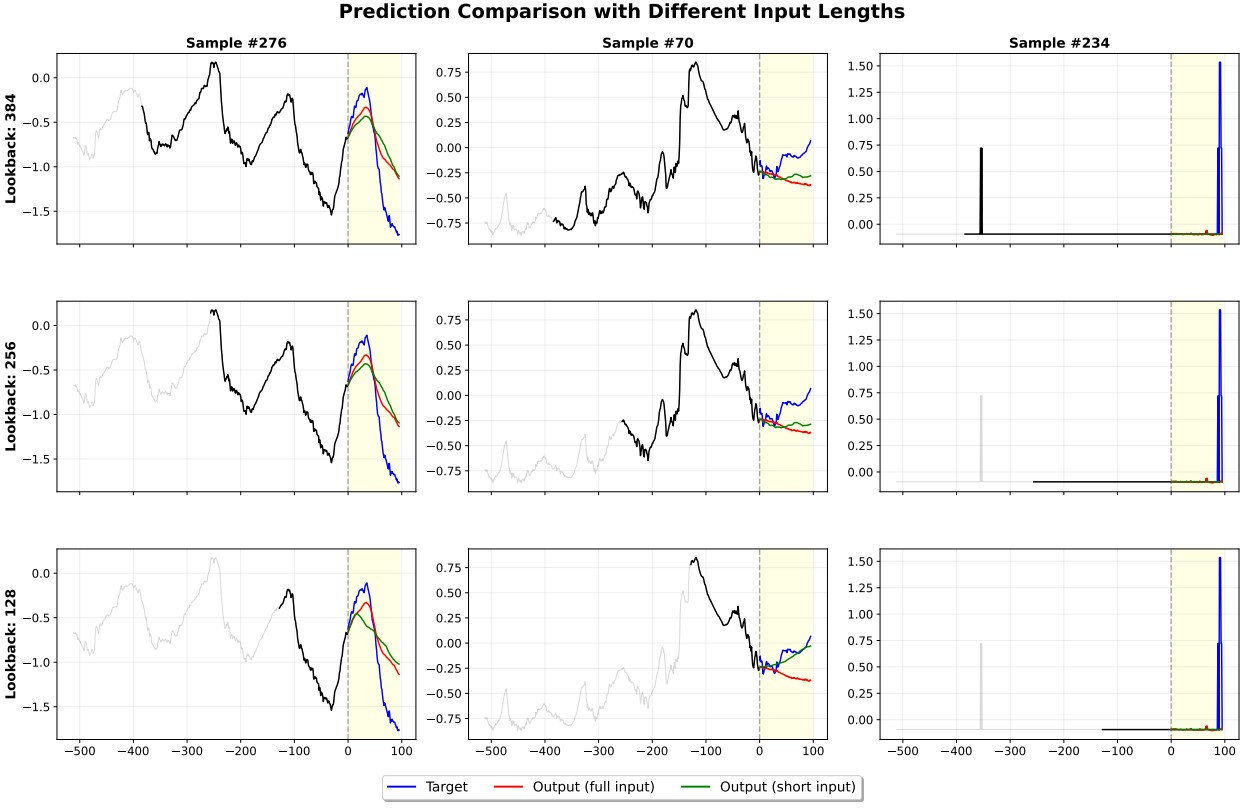

Figure 10: Super-Linear forecasting illustration with varying input lookbacks for the Weather dataset.

interpolation can increase $E_\perp(X)$ if spectral mass is shifted outside $\mathcal{S}$. Moreover, very short look-backs limit observability of long-period components, constraining achievable approximation quality irrespective of resampling.

$L_{\mathbf{input}} > 512.$   Long inputs are adaptively downsampled by minimizing

$$\text{score} = \text{entropy}(G(X_{\text{interp}})) + \lambda A_{\text{penalty}}[s],$$

subject to an energy-loss constraint.

**Benefits.** Aligns dominant frequencies with the expert span and explicitly controls energy removal.

**Limitations.**

- *Information loss*: Frequencies above the induced Nyquist limit are discarded,

$$E_{\text{lost}} = \sum_{\omega_j > \omega_{\text{Nyq}}} I_j.$$

- *Span limitation*: If dominant components lie outside $\mathcal{S}$, the irreducible term $\sqrt{E_\perp(X)}$ remains.

### B.11.2   Arbitrary Horizons via Roll-Out

For horizons exceeding the native prediction length, Super-Linear applies autoregressive roll-out.

**Benefits.** Enables arbitrary forecast lengths without architectural modification.

**Limitations.**

- *Error accumulation*: Estimation error propagates across steps.

- *Additional Overhead*: Long horizons require multiple forward passes due to roll-out. Nevertheless, Super-Linear remains highly efficient, and overall inference time is still significantly lower than attention-based TSFMs.

- *Amplified gating deviation*: The estimation term

$$\gamma \|X\|_2 \|\beta - G\|_1$$

may accumulate over long horizons.

### B.11.3 Architectural Trade-off

Super-Linear achieves flexibility through *spectral adaptation* rather than architectural expansion. This provides robustness to unseen sampling rates and controlled frequency alignment, but expressivity remains bounded by the predefined expert set $\mathcal{S}$. Unlike large attention-based TSFMs, new frequency bases are not dynamically constructed at inference time.

### B.11.4 Summary

Flexibility in Super-Linear is achieved through controlled resampling and autoregression. These mechanisms preserve robustness and interpretability but introduce smoothing bias, potential energy loss, and roll-out error accumulation. The theoretical bound highlights that performance ultimately depends on (i) residual spectral energy outside $\mathcal{S}$ and (ii) gating estimation error, both of which define the model's intrinsic flexibility limits.

## C   Theoretical Analysis

Following the *bias-complexity tradeoff* analysis (Shalev-Shwartz & Ben-David, 2014), we can decompose the error into the approximation error $\mathcal{E}_{\mathrm{app}}$, and estimation error $\mathcal{E}_{\mathrm{est}}$. The term $\mathcal{E}_{\mathrm{app}}$ represents the minimum achievable error by the Super-Linear predictor, whereas the term $\mathcal{E}_{\mathrm{est}}$ measures the difference between the approximation error and the error achieved by the ERM predictor with the given training data.

First by the triangle inequality:

$$\|Y(X) - \hat{Y}(X)\| = \left\| Y(X) - \sum_{i=1}^{N} \bar{\beta}_i F_i(X) + \sum_{i=1}^{N} \bar{\beta}_i F_i(X) - \sum_{i=1}^{N} \bar{G}_i F_i(X) \right\|$$

$$\leq \left\| Y(X) - \sum_{i=1}^{N} \bar{\beta}_i F_i(X) \right\| + \left\| \sum_{i=1}^{N} (\bar{\beta}_i - \bar{G}_i) F_i(X) \right\|$$

$$= \mathcal{E}_{\mathrm{app}} + \mathcal{E}_{\mathrm{est}}$$

**1. (Frequency) Approximation term $\mathcal{E}_{\mathrm{app}}$.**  Let $\mathcal{S} := \{\hat{\omega}_1, \ldots, \hat{\omega}_N\}$ be the set of frequencies covered by the experts. Write the Fourier expansion of the (zero-mean) target as

$$Y(X) = \sum_m \hat{X}_m \, e^{2\pi i m \cdot /T}, \qquad I_m = |\hat{X}_m|^2.$$

For now, let's assume that $X$ and $Y(X)$, share the same periodicity, formally: $I_m^X = I_m^Y = I_m$. In the following section, we will remove this assumption.

Because each $F_i$ reproduces its $\hat{\omega}_i \in \mathcal{S}$, the residual after projecting onto those frequencies is

$$\|Y(X) - R(X)\|_2^2 = \sum_{\omega_m \notin \mathcal{S}} I_m =: E_\perp(X).$$

Hence

$$\boxed{\mathcal{E}_{\mathrm{app}} \leq \sqrt{E_\perp(X)}}.$$

**2. Estimation term $\mathcal{E}_{\mathrm{est}}$.**  Assume each expert output is bounded by the energy of its frequency, i.e.

$$\|F_i(X)\|_2 \leq \gamma \sqrt{I_i}, \qquad \text{for some } \gamma > 0.$$

Using Cauchy–Schwarz and the fact that $\sum_i (\bar{\beta}_i - \bar{G}_i) = 0$ (both weight vectors sum to 1):

$$\mathcal{E}_{\mathrm{est}} = \left\| \sum_i (\bar{\beta}_i - \bar{G}_i) F_i(X) \right\|_2$$

$$\leq \left( \sum_i (\bar{\beta}_i - \bar{G}_i)^2 \right)^{1/2} \left( \sum_i \|F_i(X)\|_2^2 \right)^{1/2}$$

$$\leq \gamma \|\beta - G\|_2 \left( \sum_i I_i \right)^{1/2}$$

$$= \gamma \|\beta - G\|_2 \|X\|_2.$$

Because $\|\cdot\|_2 \leq \|\cdot\|_1$ on probability simplices,

$$\boxed{\mathcal{E}_{\mathrm{est}} \leq \gamma \|X\|_2 \|\beta - G\|_1}.$$

**3. Combined bound.** Putting the two pieces together,

$$\boxed{\; \|Y(X) - \hat{Y}(X)\| \;\leq\; \sqrt{E_\perp(X)} \;+\; \gamma \, \|X\|_2 \, \|\beta - G\|_1 \;}.$$

**Remarks.**

- $E_\perp(X)$ bounds the *(frequency)-approximation error*: the signal energy outside the span of the $N$ expert frequencies.

- The second term penalises how far the learned gating $G(X)$ is from the optimal $\beta(X)$. If training makes $\|\beta - G\|_1 \to 0$, the total error collapses to the intrinsic residual $\sqrt{E_\perp(X)}$.

To incorporate a stricter assumption on the shared periodicity. Specifically, we'll define $\delta_m$ as a frequency-dependent factor representing the relative difference in power at frequency $\omega_m$. To model a measurable difference, we assume $\delta_m$ is small, i.e., $|\delta_m| \leq \epsilon$, $\forall m$ for some small $\epsilon > 0$, and $\delta_m$ can be positive or negative (indicating amplification or attenuation of power at specific frequencies). For simplicity, we can bound the difference across all frequencies.

$$I_m^Y = (1 + \delta_m) I_m^X, \qquad \sum_{\omega_m \notin \mathcal{S}} I_m^X := E_\perp^X(X), \qquad \sum_{\omega_m \notin \mathcal{S}} I_m^Y := E_\perp(X)$$

To bound this, we use the fact that $|\delta_m| \leq \epsilon$:

$$(1 + \delta_m) I_m^X \leq (1 + \epsilon) I_m^X$$

Hence,

$$\mathcal{E}_{\text{app}} \leq \sqrt{E_\perp} \leq \sqrt{1 + \epsilon}\sqrt{E_\perp^X(X)}$$

# D  Datasets

## D.1  Pre-training Datasets

In this work, we pre-train Super-Linear using a subset of datasets from the Monash Time Series Repository (Godahewa et al., 2021) and the PEMS dataset (Liu et al., 2022), as part of the two-stage training process shown in Figure 4. Specifically, we select datasets with a minimum sequence length of 5,000 timesteps to enable sufficient downsampling. For upsampling, we cap the maximum resampling factor at $r_{\max} = 20$, which controls how much a short sequence can be stretched. For example, a sequence of length 40, when upsampled with a factor of 20, results in a sequence of length 800. The pretraining datasets are detailed in Table. 12.

## D.2  Evaluation Datasets

**LTSF**  . For evaluation, we use the standard long-horizon forecasting benchmarks: ETTm1, ETTm2, ETTh1, ETTh2, Weather, Electricity, and Traffic. We follow the same train/validation/test splits (0.6-0.2-0.2 for ETT and 0.7-0.1-0.2 for the remaining) and preprocessing protocols as in previous works (Wu et al., 2021; Zhou et al., 2021; Liu et al., 2023), including the standard scaling, which is applied independently to each channel. All the reported results represent the evaluation on the test set only. The datasets and their properties are given in Table 13.

**GIFT-Eval**  is a benchmark designed to evaluate the generalization ability of time series forecasting models across highly diverse and heterogeneous datasets. Unlike domain-specific benchmarks, it aggregates time series from a wide range of sources—including energy, finance, healthcare, traffic, and climate—covering different sampling rates, lengths, and dynamics. The goal is to test whether models trained in a universal or foundation-style setting can transfer effectively to unseen distributions without task-specific fine-tuning. By emphasizing cross-domain robustness, scalability, and zero-shot forecasting performance, GIFT-Eval serves as a more realistic and challenging measure of a model's ability to act as a general-purpose time series forecaster. A complete description of all the datasets is given in 14.

## D.3  Dataset Training Details

To prevent large datasets from dominating the training process, we limit the number of training examples per dataset to $100,000$, with a total dataset limit of $1,000,000$ examples, which translates to $1,000,000 \times (512+96)$ time-points (512 and 96 represent the lookback and horizon for training, respectively for a signel example). We also employ a uniform frequency sampling strategy: datasets are categorized by their dominant frequency (e.g., the Births dataset has a dominant frequency of $\frac{1}{7}$, corresponding to daily/weekly periodicity). This frequency label guides the resampling process to match the desired frequencies. To determine the frequency

Table 12: Datasets used for pre-training.

| Dataset | Source | Channels | Min/max channel length | Sampling rate | Frequency | Domain |
|---|---|---|---|---|---|---|
| London Smart Meters | (Godahewa et al., 2021) | 5,560 | 288/39,648 | 30min | $\frac{1}{48}$ | Energy |
| Aus. Electricity Demand | (Godahewa et al., 2021) | 5 | 230,736/232,272 | 30min | $\frac{1}{48}$ | Energy, Environmental |
| Wind Farms | (Godahewa et al., 2021) | 339 | 6,345/527,040 | minutely | $\frac{1}{1440}$ | Energy |
| KDD Cup 2018 | (Godahewa et al., 2021) | 270 | 9,504/10,920 | hourly | $\frac{1}{24}$ | Nature, Environmental |
| Weather (Monash) | (Godahewa et al., 2021) | 3,010 | 1,332/65,981 | daily | $\frac{1}{7}$ | Nature |
| Sunspot | (Godahewa et al., 2021) | 1 | 23,741 | daily | $\frac{1}{27}$ | Nature |
| Pedestrian | (Godahewa et al., 2021) | 66 | 576/96424 | hourly | $\frac{1}{24}$ | Transport, Social |
| Us Births | (Godahewa et al., 2021) | 1 | 7,305 | daily | $\frac{1}{7}$ | Nature |
| Saugeen River Flow | (Godahewa et al., 2021) | 1 | 23,741 | daily | $\frac{1}{365}$ | Nature |
| Solar Power | (Godahewa et al., 2021) | 1 | 7,397,222 | 4sec | $\frac{1}{21600}$ | Energy |
| Wind Power | (Godahewa et al., 2021) | 1 | 7,397,147 | 4sec | $\frac{1}{21600}$ | Energy |
| PEMS03 | (Liu et al., 2023) | 358 | 25,887 | 5min | $\frac{1}{288}$ | Transport |
| PEMS04 | (Liu et al., 2023) | 307 | 16,992 | 5min | $\frac{1}{288}$ | Transport |
| PEMS07 | (Liu et al., 2023) | 883 | 28,224 | 5min | $\frac{1}{288}$ | Transport |
| PEMS08 | (Liu et al., 2023) | 170 | 17,856 | 5min | $\frac{1}{288}$ | Transport |

Table 13: LTSF benchmark Datasets used for evaluation in this work.

| Dataset | Source | Channels | Min/max channel length | Sampling rate | Frequency | Domain | Usage |
|---|---|---|---|---|---|---|---|
| ETTh1 | (Wu et al., 2021; Zhou et al., 2021) | 7 | 17,420 | hourly | - | Energy | Evaluation |
| ETTh2 | (Wu et al., 2021; Zhou et al., 2021) | 7 | 17,420 | hourly | - | Energy | Evaluation |
| ETTm1 | (Wu et al., 2021; Zhou et al., 2021) | 7 | 69,680 | 15min | - | Energy | Evaluation |
| ETTm2 | (Wu et al., 2021; Zhou et al., 2021) | 7 | 69,680 | 15min | - | Energy | Evaluation |
| Electricity | (Wu et al., 2021; Zhou et al., 2021) | 321 | 26,304 | hourly | - | Energy | Evaluation |
| Traffic | (Wu et al., 2021; Zhou et al., 2021) | 862 | 17,544 | hourly | - | Transport, Environmental | Evaluation |
| Weather | (Wu et al., 2021; Zhou et al., 2021) | 21 | 52,696 | 10min | - | Nature | Evaluation |

Table 14: Individual statistics of GIFT-Eval benchmark across all datasets as given in (Aksu et al., 2024).

| Dataset | Source | Domain | Samling Rate. | #Channels | Channels Length | | | #Obs | Target Var | Short-term | | Med-term | | Long-term | |
|---|---|---|---|---|---|---|---|---|---|---|---|---|---|---|---|
| | | | | | Avg | Min | Max | | | Len | Win | Len | Win | Len | Win |
| Jena Weather | (Zhou et al., 2021) | Nature | 10min | 1 | 52,704 | 52,704 | 52,704 | 52,704 | 21 | 48 | 20 | 480 | 11 | 720 | 8 |
| Jena Weather | (Zhou et al., 2021) | Nature | hourly | 1 | 8,784 | 8,784 | 8,784 | 8,784 | 21 | 48 | 19 | 480 | 2 | 720 | 2 |
| Jena Weather | (Zhou et al., 2021) | Nature | daily | 1 | 366 | 366 | 366 | 366 | 21 | 30 | 2 | - | - | - | - |
| BizITObs - App | Palaskar et al. (2024) | Web/CloudOps | 10sec | 1 | 8,834 | 8,834 | 8,834 | 8,834 | 2 | 60 | 15 | 600 | 2 | 900 | 1 |
| BizITObs - Service | Palaskar et al. (2024) | Web/CloudOps | 10sec | 21 | 8,835 | 8,835 | 8,835 | 185,535 | 2 | 60 | 15 | 600 | 2 | 900 | 1 |
| BizITObs - L2C | Palaskar et al. (2024) | Web/CloudOps | 5min | 1 | 31,968 | 31,968 | 31,968 | 31,968 | 7 | 48 | 20 | 480 | 7 | 720 | 5 |
| BizITObs - L2C | Palaskar et al. (2024) | Web/CloudOps | hourly | 1 | 2,664 | 2,664 | 2,664 | 2,664 | 7 | 48 | 6 | 480 | 1 | 720 | 1 |
| Bitbrains - Fast | Shen et al. (2015) | Web/CloudOps | 5min | 1,250 | 8,640 | 8,640 | 8,640 | 10,800,000 | 2 | 48 | 18 | 480 | 2 | 720 | 2 |
| Bitbrains - Fast | Shen et al. (2015) | Web/CloudOps | hourly | 1,250 | 721 | 721 | 721 | 901,250 | 2 | 48 | 2 | - | - | - | - |
| Bitbrains - rnd | Shen et al. (2015) | Web/CloudOps | 5min | 500 | 8,640 | 8,640 | 8,640 | 4,320,000 | 2 | 48 | 18 | 480 | 2 | 720 | 2 |
| Bitbrains - rnd | Shen et al. (2015) | Web/CloudOps | hourly | 500 | 720 | 720 | 720 | 360,000 | 2 | 48 | 2 | - | - | - | - |
| Restaurant | (Aksu et al., 2024) | Sales | daily | 807 | 358 | 67 | 478 | 289,303 | 1 | 30 | 1 | - | - | - | - |
| ETT1 | (Zhou et al., 2021) | Energy | 15min | 1 | 69,680 | 69,680 | 69,680 | 69,680 | 7 | 48 | 20 | 480 | 15 | 720 | 10 |
| ETT1 | (Zhou et al., 2021) | Energy | hourly | 1 | 17,420 | 17,420 | 17,420 | 17,420 | 7 | 48 | 20 | 480 | 4 | 720 | 3 |
| ETT1 | (Zhou et al., 2021) | Energy | daily | 1 | 725 | 725 | 725 | 725 | 7 | 30 | 3 | - | - | - | - |
| ETT1 | (Zhou et al., 2021) | Energy | weekly-THU | 1 | 103 | 103 | 103 | 103 | 7 | 8 | 2 | - | - | - | - |
| ETT2 | (Zhou et al., 2021) | Energy | 15min | 1 | 69,680 | 69,680 | 69,680 | 69,680 | 7 | 48 | 20 | 480 | 15 | 720 | 10 |
| ETT2 | (Zhou et al., 2021) | Energy | hourly | 1 | 17,420 | 17,420 | 17,420 | 17,420 | 7 | 48 | 20 | 480 | 4 | 720 | 3 |
| ETT2 | (Zhou et al., 2021) | Energy | daily | 1 | 725 | 725 | 725 | 725 | 7 | 30 | 3 | - | - | - | - |
| ETT2 | (Zhou et al., 2021) | Energy | weekly-THU | 1 | 103 | 103 | 103 | 103 | 7 | 8 | 2 | - | - | - | - |
| Loop Seattle | (Wang et al., 2023) | Transport | 5min | 323 | 105,120 | 105,120 | 105,120 | 33,953,760 | 1 | 48 | 20 | 480 | 20 | 720 | 15 |
| Loop Seattle | (Wang et al., 2023) | Transport | hourly | 323 | 8,760 | 8,760 | 8,760 | 2,829,480 | 1 | 48 | 19 | 480 | 2 | 720 | 2 |
| Loop Seattle | (Wang et al., 2023) | Transport | daily | 323 | 365 | 365 | 365 | 117,895 | 1 | 30 | 2 | - | - | - | - |
| SZ-Taxi | (Wang et al., 2023) | Transport | 15min | 156 | 2,976 | 2,976 | 2,976 | 464,256 | 1 | 48 | 7 | 480 | 1 | 720 | 1 |
| SZ-Taxi | (Wang et al., 2023) | Transport | hourly | 156 | 744 | 744 | 744 | 116,064 | 1 | 48 | 2 | - | - | - | - |
| M_DENSE | (Wang et al., 2023) | Transport | hourly | 30 | 17,520 | 17,520 | 17,520 | 525,600 | 1 | 48 | 20 | 480 | 4 | 720 | 3 |
| M_DENSE | (Wang et al., 2023) | Transport | daily | 30 | 730 | 730 | 730 | 21,900 | 1 | 30 | 3 | - | - | - | - |
| Solar | (Lai et al., 2018) | Energy | 10min | 137 | 52,560 | 52,560 | 52,560 | 7,200,720 | 1 | 48 | 20 | 480 | 11 | 720 | 8 |
| Solar | (Lai et al., 2018) | Energy | hourly | 137 | 8,760 | 8,760 | 8,760 | 1,200,120 | 1 | 48 | 19 | 480 | 2 | 720 | 2 |
| Solar | (Lai et al., 2018) | Energy | daily | 137 | 365 | 365 | 365 | 50,005 | 1 | 30 | 2 | - | - | - | - |
| Solar | (Lai et al., 2018) | Energy | weekly-FRI | 137 | 52 | 52 | 52 | 7,124 | 1 | 8 | 1 | - | - | - | - |
| Hierarchical Sales | (Mancuso et al., 2021) | Sales | daily | 118 | 1,825 | 1,825 | 1,825 | 215,350 | 1 | 30 | 7 | - | - | - | - |
| Hierarchical Sales | (Mancuso et al., 2021) | Sales | weekly-WED | 118 | 260 | 260 | 260 | 30,680 | 1 | 8 | 4 | - | - | - | - |
| M4 Yearly | Godahewa et al. (2021) | Econ/Fin | yearly-DEC | 22,974 | 37 | 19 | 284 | 845,109 | 1 | 6 | 1 | - | - | - | - |
| M4 Quarterly | Godahewa et al. (2021) | Econ/Fin | quarterly-DEC | 24,000 | 100 | 24 | 874 | 2,406,108 | 1 | 8 | 1 | - | - | - | - |
| M4 Monthly | Godahewa et al. (2021) | Econ/Fin | monthly | 48,000 | 234 | 60 | 2,812 | 11,246,411 | 1 | 18 | 1 | - | - | - | - |
| M4 Weekly | Godahewa et al. (2021) | Econ/Fin | weekly-SUN | 359 | 1,035 | 93 | 2,610 | 371,579 | 1 | 13 | 1 | - | - | - | - |
| M4 Daily | Godahewa et al. (2021) | Econ/Fin | daily | 4,227 | 2,371 | 107 | 9,933 | 10,023,836 | 1 | 14 | 1 | - | - | - | - |
| M4 Hourly | Godahewa et al. (2021) | Econ/Fin | hourly | 414 | 902 | 748 | 1,008 | 373,372 | 1 | 48 | 2 | - | - | - | - |
| Hospital | Godahewa et al. (2021) | Healthcare | monthly | 767 | 84 | 84 | 84 | 64,428 | 1 | 12 | 1 | - | - | - | - |
| COVID Deaths | Godahewa et al. (2021) | Healthcare | daily | 266 | 212 | 212 | 212 | 56,392 | 1 | 30 | 1 | - | - | - | - |
| US Births | Godahewa et al. (2021) | Healthcare | daily | 1 | 7,305 | 7,305 | 7,305 | 7,305 | 1 | 30 | 20 | - | - | - | - |
| US Births | Godahewa et al. (2021) | Healthcare | weekly-TUE | 1 | 1,043 | 1,043 | 1,043 | 1,043 | 1 | 8 | 14 | - | - | - | - |
| US Births | Godahewa et al. (2021) | Healthcare | monthly | 1 | 240 | 240 | 240 | 240 | 1 | 12 | 2 | - | - | - | - |
| Saugeen | Godahewa et al. (2021) | Nature | daily | 1 | 23,741 | 23,741 | 23,741 | 23,741 | 1 | 30 | 20 | - | - | - | - |
| Saugeen | Godahewa et al. (2021) | Nature | weekly-THU | 1 | 3,391 | 3,391 | 3,391 | 3,391 | 1 | 8 | 20 | - | - | - | - |
| Saugeen | Godahewa et al. (2021) | Nature | monthly | 1 | 780 | 780 | 780 | 780 | 1 | 12 | 7 | - | - | - | - |
| Temp. Rain | Godahewa et al. (2021) | Nature | daily | 32,072 | 725 | 725 | 725 | 780 | 1 | 30 | 3 | - | - | - | - |
| KDD Cup 2018 | Godahewa et al. (2021) | Nature | hourly | 270 | 10,898 | 9,504 | 10,920 | 2,942,364 | 1 | 48 | 20 | 480 | 2 | 720 | 2 |
| KDD Cup 2018 | Godahewa et al. (2021) | Nature | daily | 270 | 455 | 396 | 455 | 122,791 | 1 | 30 | 2 | - | - | - | - |
| Car Parts | Godahewa et al. (2021) | Sales | monthly | 2,674 | 51 | 51 | 51 | 136,374 | 1 | 12 | 1 | - | - | - | - |
| Electricity | (Asuncion et al., 2007) | Energy | 15min | 370 | 140,256 | 140,256 | 140,256 | 51,894,720 | 1 | 48 | 20 | 480 | 20 | 720 | 20 |
| Electricity | (Asuncion et al., 2007) | Energy | hourly | 370 | 35,064 | 35,064 | 35,064 | 12,973,680 | 1 | 48 | 20 | 480 | 8 | 720 | 5 |
| Electricity | (Asuncion et al., 2007) | Energy | daily | 370 | 1,461 | 1,461 | 1,461 | 540,570 | 1 | 30 | 5 | - | - | - | - |
| Electricity | (Asuncion et al., 2007) | Energy | weekly-FRI | 370 | 208 | 208 | 208 | 76,960 | 1 | 8 | 3 | - | - | - | - |

of each dataset, we use the sampling rate and a manual spectrum analysis, the analyzed frequencies are also detailed in Table 12.

The training complexity of Super-Linear, consists of two stages with runtimes $T_1$ and $T_2$ (see Fig. 4), for the stage 1 expert training and the stage 2 router + complementary experts training, respectively. Here, $T_1 = \frac{T_{expert}}{c}$, where $T_{\text{expert}}$ is the training time for a single expert and $c$ is the number of available GPUs.

On our system (RTX3090), we measure:

$T_2 \approx 1.17$hours, and $T_{expert} \approx 0.72$hours

**Data Resampling** To ensure frequency diversity across the pre-trained datasets, we transform each dataset to span a wider range of frequencies, as illustrated in 11. For each dataset, we sample a sequence and apply a *scaling factor* that either expands or compresses the signal, effectively shifting its dominant frequency. Since every dataset has a known dominant frequency, we can systematically rescale it to obtain sequences representing different frequencies.

For example, a dataset sampled at an hourly or daily rate with a dominant frequency of $\frac{1}{24}$ can be downsampled by a factor of $\frac{7}{24}$, yielding a sequence of length $\frac{7}{24}$ of the original. The same procedure applies for upsampling, where signals are expanded using a scaling factor greater than 1. The maximal scaling factor used is 20. We use linear interpolation as the resampling operator.

**Common Frequencies** In this work, we select multiple frequencies corresponding to natural sampling rates and train a separate linear layer expert for each one, forming the Super-Linear model. The chosen frequencies and their respective sampling rates are illustrated in Figure 11.

Figure 11: The common frequencies utilized by Super-Linear and their corresponding sampling rates. Each frequency expert in Super-Linear is learned with a frequency depicted in this figure, and each frequency is associated with one or more natural sampling rates.

During training, we uniformly sample from the pool of target frequencies (see Appendix D.3), independently of the dataset source. This reflects our focus on frequency diversity rather than dataset diversity. For each dataset, we use the first 80% of its timeline for training and reserve the remaining 20% for validation. Since datasets vary in channel dimensionality and length, we adopt channel-independent (i.e., univariate) training, treating each channel as a separate time series.

# E  Benchmark Models

## E.1  Pretrained Models

**Timer-XL** (Liu et al., 2024c) is a decoder-only Transformer that models multivariate time series as flattened sequences of patch tokens, enabling causal next-token prediction across both time and variables. Its core component, TimeAttention, uses a Kronecker product of temporal and variable masks to enforce causal structure and capture inter-variable dependencies, while maintaining permutation-equivalence over input variables.

**Time-MoE** (Shi et al., 2024) is a decoder-only Transformer architecture for time series forecasting that introduces sparse mixture-of-experts (MoE) layers to scale model capacity while maintaining efficiency, dynamically activating only a subset of expert networks per input token. It employs point-wise tokenization for flexible sequence handling, rotary positional embeddings for better extrapolation, and a multi-resolution forecasting head to support arbitrary forecast horizons.

**Moirai** (Woo et al., 2024) is a masked encoder-based Transformer designed for universal time series forecasting, featuring multi-patch size input/output projections to handle multiple sampling frequencies and Any-variate Attention to flexibly model arbitrary multivariate sequences. It outputs parameters for a flexible mixture of parametric distributions, enabling probabilistic forecasting across diverse domains, variate dimensionalities, and data distributions.

**TimesFM** (Das et al., 2024) model is a decoder-only Transformer architecture for time-series forecasting that uses input patching to efficiently represent and process long sequences, inspired by tokenization in language models. It is pretrained on a large and diverse corpus of real and synthetic time-series data, enabling accurate zero-shot forecasts across different domains, time granularities, and forecast horizons.

**Chronos** (Ansari et al., 2024) is an adaptation of language models for time series forecasting, where time series data is tokenized through scaling and quantization into discrete bins, allowing standard language models to be used for probabilistic forecasting. It trains models on a large set of time series, using data augmentation techniques like TSMixup and KernelSynth, and performs zero-shot forecasting without the need for task-specific adjustments or architecture changes.

**ROSE** (Wang et al.) a lightweight general time series forecasting model designed for multi-domain data. Unlike other foundation models that mainly rely on scaling datasets and model sizes, ROSE tackles two key challenges: learning unified representations across heterogeneous domains and enabling adaptive transfer to domain-specific tasks. To achieve this, the authors propose Decomposed Frequency Learning, which disentangles complex temporal patterns by applying frequency-based masking and reconstruction, leading to generalized representations, and the Time Series Register (TS-Register), which stores domain-specific representations during pre-training and adaptively selects relevant ones for downstream tasks via fine-tuning with low-rank updates.

**TTM** (Ekambaram et al., 2024) Multi-level Tiny Time Mixers are lightweight pre-trained models for multivariate time-series forecasting, designed to rival or outperform large foundation models while using only 1M parameters. Built on the efficient TSMixer architecture, TTM replaces costly Transformer self-attention with MLP-Mixer blocks and gated attention, enabling fast pre-training and inference. To handle heterogeneous, multi-resolution datasets, TTM introduces adaptive patching (AP), diverse resolution sampling (DRS), and resolution prefix tuning (RPT), which improve robustness across domains and temporal scales. Its multi-level modeling strategy first pre-trains channel-independent representations, then fine-tunes with channel mixing to capture correlations across targets and exogenous signals—capabilities often missing in larger models.

**Sundial** (Liu et al., 2025b) is a family of scalable time series foundation models that directly model continuous sequences using TimeFlow Loss, a flow-matching objective that enables Transformers to generate multiple probable forecasts without discrete tokenization or parametric densities. With minimal Transformer adaptations and pre-training on TimeBench (a trillion time points), Sundial achieves state-of-the-art zero-shot results on major benchmarks while delivering fast, flexible, and reliable generative forecasting.

**VisionTS** (Chen et al., 2024) repurposes a pre-trained visual masked autoencoder (MAE) for time series forecasting by converting time series into images and framing forecasting as masked patch reconstruction. This simple reformulation enables strong zero-shot performance and near state-of-the-art results with minimal fine-tuning, showing visual models can serve as effective foundation models for time series.

**TiRex** (Auer et al., 2025) TiRex is a zero-shot forecasting model built on xLSTM, a modern LSTM variant that combines strong state-tracking with competitive in-context learning. Its design is enhanced by Contiguous Patch Masking, which stabilizes long-horizon autoregressive predictions, and by tailored data augmentations that improve robustness. This architecture enables TiRex to outperform much larger transformer-based models on standard benchmarks.

### E.2 In-Domain Models

**UniTST** (Liu et al., 2024a) model employs a unified attention mechanism that simultaneously captures both inter-variate and intra-variate dependencies by flattening patches from different variates into a unified sequence. To reduce memory complexity, it introduces a dispatcher mechanism, which aggregates dependencies across tokens and distributes the information efficiently, ensuring effective modeling of time-series data with large variates.

**CycleNet** (Lin et al., 2024a) is designed to explicitly model periodic patterns in time series data using the Residual Cycle Forecasting (RCF) technique, which leverages learnable recurrent cycles to capture inherent cyclic components. The model then predicts the residual components of the cycles using a simple backbone, either a linear or shallow MLP model, providing an efficient and powerful solution for long-term time series forecasting.

**iTransformer** (Liu et al., 2023) model design independently embeds each time series as a variate token, applying self-attention to capture multivariate correlations between these tokens. The model then utilizes feed-forward networks to learn series representations, eliminating the need for complex encoder-decoder architectures while improving efficiency and forecasting accuracy.

**TimesNet**, (Wu et al., 2022) a deep learning framework designed to capture intricate temporal variations in time series data by leveraging the multi-periodicity inherent in real-world time series. It introduces several key innovations and methods for better handling complex temporal patterns, such as intraperiod-variation (short-term variations within a period) and interperiod-variation (long-term variations between different periods).

**PatchTST** (Nie et al., 2023) is a Transformer-based architecture designed for multivariate time series forecasting, incorporating two key innovations: patching and channel-independence. Patching divides time series into non-overlapping subseries to capture local semantic information, while channel-independence processes each univariate time series independently to reduce computational complexity and enhance forecasting accuracy.

**Autoformer** (Wu et al., 2021) is a time series forecasting model that incorporates a decomposition architecture to handle long-term temporal patterns. It introduces an Auto-Correlation mechanism that efficiently discovers period-based dependencies and aggregates similar sub-series, significantly improving both information utilization and computational efficiency for long-term forecasting.

**FreqMoE** (Liu, 2025) integrates a Frequency Decomposition Mixture-of-Experts (MoE) module, where expert networks focus on distinct frequency bands of the time series data, with a gating network dynamically adjusting the contribution of each expert based on the frequency magnitude. The model then utilizes residual-connected frequency domain prediction blocks to iteratively refine predictions, effectively capturing complex temporal and frequency patterns for accurate time series forecasting.

**MoLE**, (Ni et al., 2024) The Mixture-of-Linear-Experts model augments linear-centric time series forecasting models by training multiple linear models (experts) and a router model that adaptively mixes their outputs based on temporal patterns. Using an embedding of the first timestamp, the router assigns weights to each expert, allowing them to specialize in different temporal periods, thus improving forecasting accuracy while maintaining the simplicity of linear models.

**MTLinear** (Nochumsohn et al., 202t) for multivariate time series forecasting groups similar variates based on their Pearson Correlation Coefficient, assigning each group a separate linear module. This design leverages multi-task learning principles to address gradient conflicts and balance the contribution of variates during training, improving forecasting accuracy and efficiency.

**DLinear**, (Zeng et al., 2023) a linear based model that first decomposes the input time series into trend and seasonal components using a moving average kernel. It then applies separate one-layer linear models to each component and combines their outputs to make the final prediction.

**DUET**, (Qiu et al., 2025) is a multivariate time series forecasting framework that uses dual clustering to handle heterogeneity in both time and channels. It applies a Temporal Clustering Module (TCM) to group the series into fine-grained distribution clusters and assigns specialized pattern extractors to each, capturing diverse temporal behaviors caused by distribution shifts. It then uses a Channel Clustering Module (CCM) with soft clustering in the frequency domain (via metric learning + sparsification) to flexibly model complex, noisy inter-channel relationships, followed by a masked-attention fusion step that integrates both clustering results for stronger predictions.

**FEDformer**, (Zhou et al., 2022) integrates seasonal-trend decomposition into a Transformer to capture the global time series profile (trend + seasonality), overcoming standard Transformer's weakness in preserving overall distribution for long-term forecasting. It replaces self- and cross-attention with frequency-enhanced blocks (using random sparse Fourier/Wavelet components) to better model structures in the frequency domain while reducing complexity from quadratic to linear.

**Time-VLM**, (Zhong et al., 2025) is a multimodal framework that unifies time series with vision and text by leveraging pre-trained vision-language models (VLMs) to improve forecasting, especially in few-shot and zero-shot scenarios. It combines three learners: (1) a retrieval-augmented temporal learner for hierarchical feature extraction from raw sequences, (2) a vision-augmented learner that adaptively converts time series into multi-scale images using convolutions, frequency, and periodic encoding, and (3) a text-augmented learner that generates rich contextual descriptions and statistics, enabling cross-modal fusion within the VLM to produce more accurate and generalizable predictions.

**Time-LLM**, (Jin et al., 2023) reprograms frozen LLMs for time series forecasting by patching/normalizing input series, then aligning patch embeddings to the LLM's token space via learned text prototypes and cross-attention, without touching the backbone. It enriches reasoning with Prompt-as-Prefix—natural language prefixes containing domain context, task instructions, and statistics (trends, lags, min/max/median)—before projecting the LLM's output embeddings to final forecasts, enabling strong few-/zero-shot generalization.

**TimeCMA**, (Liu et al., 2025a) uses a dual-modality encoding approach: one branch extracts disentangled (but initially weak) embeddings directly from multivariate time series, while the other wraps the same series into carefully designed textual prompts and feeds them to a frozen LLM to obtain entangled yet robust prompt embeddings rich in temporal information. It then performs cross-modality alignment by retrieving the strongest disentangled yet enhanced time series components from the LLM's prompt embeddings via channel-wise similarity, aggregates them into the original time series embeddings, and uses only the last token embedding of each prompt (stored offline for efficiency) to enable fast, high-quality multivariate forecasting.

# F   Super-Linear Frequency Expert Weights

As shown in Figure 12, the frequency experts capture distinct, well-defined patterns associated with the corresponding periodic lag. In addition, the complementary experts reveal additional structure that the model did not learn during stage 1 training. For example, `comp_11` and `comp_8` exhibit activations similar to those of the frequency experts, suggesting they encode complementary frequency information. In contrast, experts such as `comp_0` and `comp_5` display more arbitrary behavior, which appears to compensate for patterns that the frequency experts may struggle to extract.

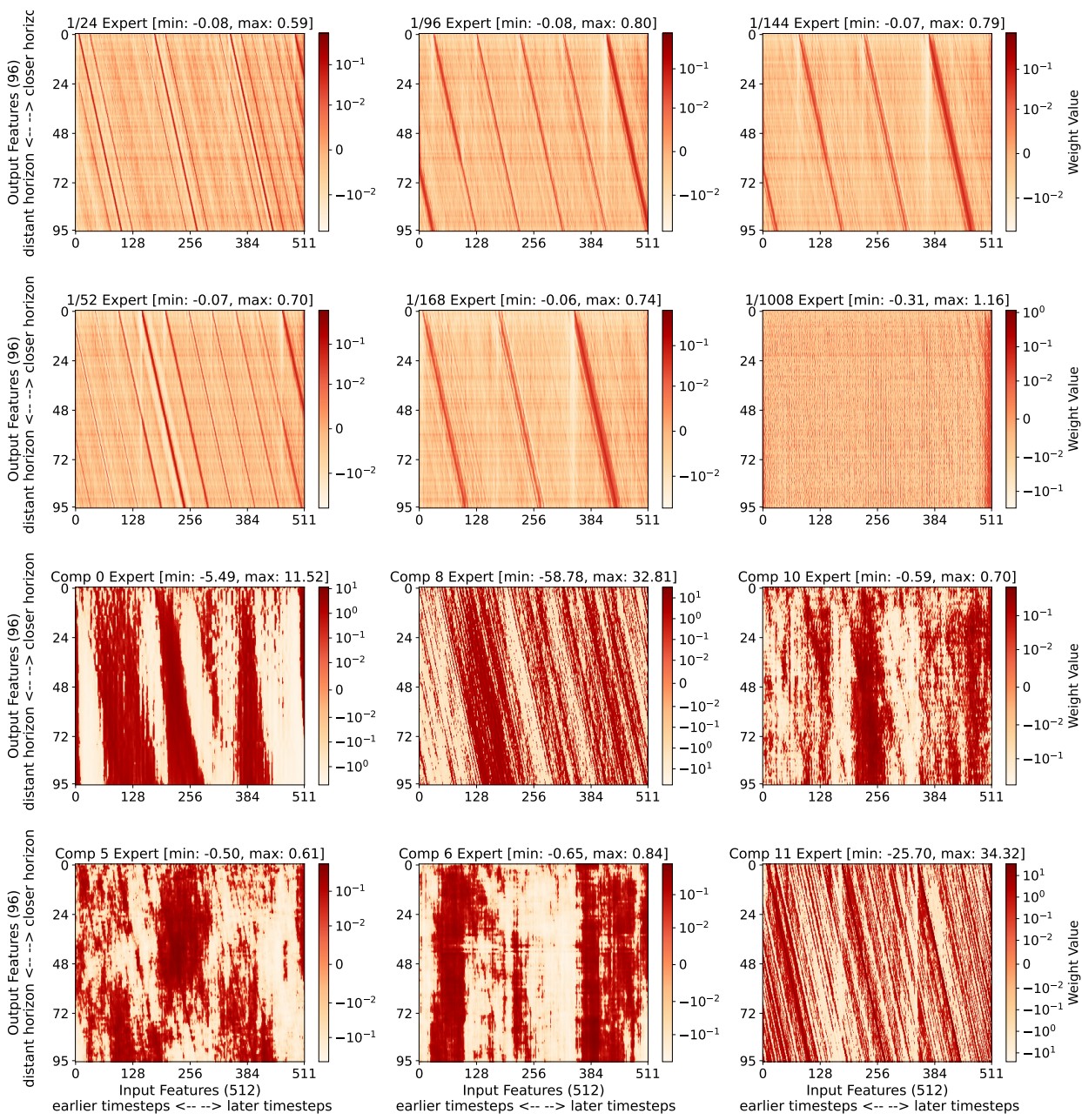

Figure 12: Visualization of the Super-Linear frequency and complementary expert weights.

# G    Hyperparameters

In this section we present the hyper parameter for the pretrained ZS Super-Linear 15 and in-domain (FS) 16.

# H    Limitations

Despite its efficiency and strong performance, Super-Linear has several limitations. First, its reliance on linear representations may restrict its ability to capture complex temporal dependencies or nonlinear patterns that are common in many real-world time series. While the frequency-specialized design helps mitigate this

Table 15: Description of hyperparameters for pretrained (ZS) Super-Linear configuration.

| Hyperparameter | Description | Values |
|---|---|---|
| Learning Rate | Controls the step size in gradient descent | 0.1 |
| Batch Size | Number of training examples used in one iteration | 512 |
| Channel Organization | Processing input data as either univariate (channel independent) or multivariate during training | Channel Independent |
| Number of Epochs | Number of times the entire dataset is passed through the model | 30 |
| Patience | Number of epochs without improvement before early stopping | 5 |
| Learning Rate Adjustment (decay) | Strategy for adapting the learning rate during training | $0.9^{epoch-3}$ if $epoch > 3$ else 1 |
| Top-k Experts | Number of selected experts in training | 12 |
| Dataset Max Size | Maximum number of samples considered for training from each dataset | $100,000$ |
| Frequency Experts $N_f$ | Number of different frequency experts | 37 |
| Complementary Experts $N_c$ | Additional experts used alongside the frequency ones ones | 12 |
| Optimizer | Algorithm used for model parameter optimization | Adam |
| Gate Weight Dimension $M$ | Dimension proportional to spectrum length | $2,500$ |
| Lookback | Number of past timesteps used as input for prediction | 512 |
| Forecast Horizon | Future timesteps predicted by the model during training only | 96 |
| Gate Noise Std | Standard deviation of noise added to gating mechanism | 0.1 |
| Maximal Energy Loss $E_{maxloss}$ | The maximum allowed fraction of normalized energy loss during the resampling process | 0.2 |
| Scales $S = \{s_1, s_2, ...\}$ | The resampling scale factor for the interpolation of the *Long Lookback Resample Search Algorithm* | $\{2,4,6\}$ |
| Energy Loss Penalty Scaler $\lambda$ | A scaler to control the incurred energy loss penalty of the *Long Lookback Resample Search Algorithm* | 2 |

Table 16: Description of hyperparameters for in-domain (FS) Super-Linear configuration.

| Hyperparameter | Description | Values |
|---|---|---|
| Learning Rate | Controls the step size in gradient descent | 0.05 |
| Batch Size | Number of training examples used in one iteration | 32 |
| Channel Organization | Processing input data as either univariate (channel independent) or multivariate during training | Multivariate |
| Number of Epochs | Number of times the entire dataset is passed through the model | 30 |
| Patience | Number of epochs without improvement before early stopping | 5 |
| Learning Rate Adjustment (decay) | Strategy for adapting the learning rate during training | none |
| Top-k Experts | Number of selected experts in training | $6, 8, 10, 12, 20$ |
| Dataset Max Size | Maximum number of samples considered for training from each dataset | none |
| Frequency Experts $N_f$ | Number of different frequency experts | 37 |
| Complementary Experts $N_c$ | Additional experts used alongside the frequency ones ones | 10 |
| Optimizer | Algorithm used for model parameter optimization | Adam |
| Gate Weight Dimension $M$ | Dimension proportional to spectrum length | $2,500$ |
| Lookback | Number of past timesteps used as input for prediction | 512 |
| Forecast Horizon | Future timesteps predicted by the model during training only | 96 |
| Gate Noise Std | Standard deviation of noise added to gating mechanism | 0.1 |

to some extent, it may still fall short in scenarios where nonlinear interactions between frequencies are crucial. Second, although Super-Linear can process relatively long input sequences due to its lightweight architecture, its context modeling is limited by 512 lookback steps, which may hinder performance on tasks that require reasoning over very long lookback contexts. Finally, the model may struggle with time series that exhibit uncommon or transient frequency components not well-aligned with the predefined frequencies in 11. Expert specialization relies on fixed frequency filters, which may struggle to represent rare or shifting frequency patterns, potentially hindering generalization in such scenarios. While complementary layers aim to bridge information gaps—particularly for uncommon frequencies—this limitation may still persist.

### H.1 Training–Evaluation Overlap Analysis

There is limited overlap between the pretraining corpus and the GIFT-Eval benchmark. Specifically, the *Saugeen*, *Births*, and *KDD* datasets appear in both collections. Regarding the *Solar* dataset, it was not included in the training corpus; it was mistakenly listed in Table 12 of the original manuscript and was removed in the revised version.

Importantly, several competing methods, including *MOIRAI*, *TTM-R1/R2*, *Time-MoE*, *Chronos-Bolt/Base*, *TimesFM 1.0/2.0*, and likely *Sundial*, are trained on LOTSA Woo et al. (2024) or subsets of it, which also contains datasets used in GIFT-Eval. This provides important context when assessing the fairness of comparison.

To quantify the effect of overlap, we trained an additional version of Super-Linear that excludes all overlapping datasets (denoted as *no leak*). The results are reported below.

Table 17: LTSF benchmark results (MSE).

| Model | ETTm1 | ETTm2 | ETTh2 | ETTh1 | Traffic | Weather | Electricity |
|---|---|---|---|---|---|---|---|
| Super-Linear | 0.317 | 0.179 | 0.279 | 0.369 | 0.414 | 0.159 | 0.141 |
| Super-Linear (no leak) | 0.316 | 0.177 | 0.277 | 0.371 | 0.412 | 0.162 | 0.148 |

Table 18: GIFT-Eval benchmark results (MASE).

| Model | GIFT-Eval |
|---|---|
| Super-Linear | 0.857 |
| Super-Linear (no leak) | 0.863 |

The differences are marginal and mixed across benchmarks, indicating that dataset overlap has minimal impact on performance. These findings further support the conclusion that Super-Linear benefits primarily from frequency diversity rather than sheer data abundance.

# I  Case Study Visualization

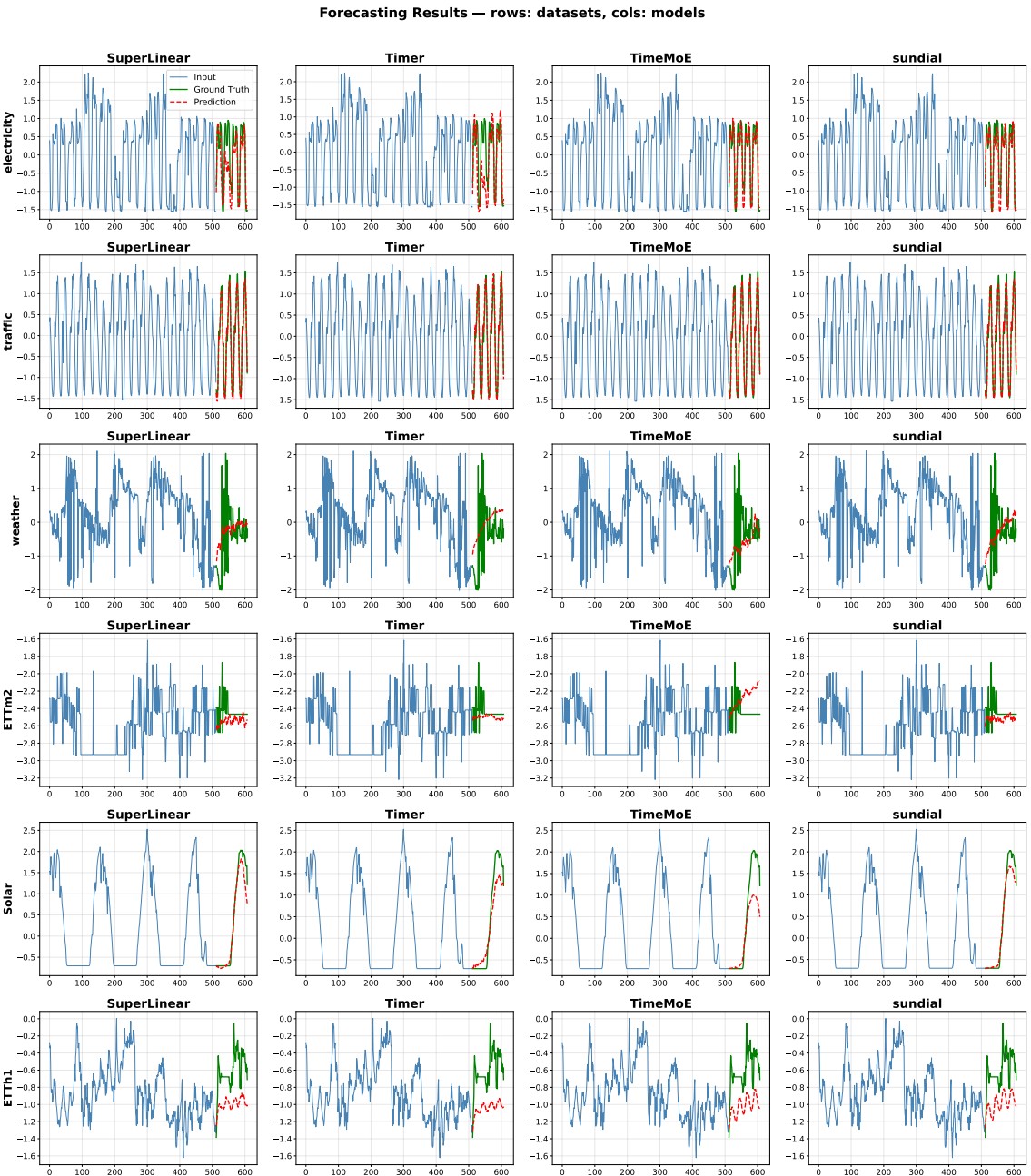

Figure 13: Forecasting results across datasets and models. Rows correspond to datasets (Electricity, Traffic, Weather, ETTm2, Solar, ETTh1), columns correspond to models (SuperLinear, Timer, TimeMoE, Sundial). Blue: input context, green: ground truth, red dashed: prediction.

