# OpenReview forum: "Super-Linear: A Lightweight Pretrained Mixture of Linear Experts for Time Series Forecasting"
_TMLR — Accepted by TMLR_

### Review · Reviewer_P2Up · 2026-02-17

**Summary Of Contributions:**

This paper presents a new framework for lightweight pretrained time series forecasting with MoE. Generally, the paper is well-written and easy to follow. In addition, the method is sound. Further, the experiments show the effectiveness of the proposals.

**Audience:**

Yes

**Audience Explanation:**

This paper may motivate the time series community, which is a part of TMLR audience.

**Claims And Evidence:**

Yes

**Claims Explanation:**

The claims are supported by the experiments and/or claims.

**Requested Changes:**

1. It would be better to include the classic deep learning based baselines for comparison, enabling fairness, such as

[1]. A Time Series is Worth 64 Words: Long-term Forecasting with Transformers, ICLR 2023.

[2]. iTransformer: Inverted Transformers Are Effective for Time Series Forecasting, ICLR 2024.

[3]. FEDformer: Frequency Enhanced Decomposed Transformer for Long-term Series Forecasting, ICML 2022.

[4]. DUET: Dual clustering enhanced multivariate time series forecasting, KDD 2025.

2. More LLM-empowered baselines are required as follows.

[1]. Time-LLM: Time Series Forecasting by Reprogramming Large Language Models, ICLR 2024.

[2]. Time-VLM: Exploring Multimodal Vision-Language Models for Augmented Time Series Forecasting, ICML 2025.

[3]. TimeCMA: Towards LLM-Empowered Multivariate Time Series Forecasting via Cross-Modality Alignment, AAAI 2025.

3. It would be better to include a case study to intuitively show the effectiveness of the proposals.

---

> ### Author Response · Authors · 2026-03-08
>
> We thank the reviewer for the positive assessment of our work and the constructive feedback. We appreciate the suggestions regarding additional baselines and further qualitative analysis. In the revised version, we expanded the experimental evaluation and clarified the comparisons where appropriate. Below we address the reviewer’s comments in detail.
>
> ---
>
> >It would be better to include the classic deep learning based baselines for comparison, enabling fairness, such as...
>
> We thank the reviewer for the suggestion. Classical deep learning baselines such as iTransformer and PatchTST are already included in Table~4 (Appendix, full-shot comparison), and we have additionally incorporated FEDformer and DUET to further improve fairness. While reproducing DUET, we observed discrepancies with the originally reported results, likely due to differences between its official repository and the standard LTSF framework (e.g., TSLib) used by other methods. To ensure consistency, we re-implemented DUET within the same LTSF codebase and evaluation protocol, using the reported architecture and hyperparameters, and report these reproduced results for a fair comparison.
>
> ---
>
> >More LLM-empowered baselines are required as follows....
>
> In Tab.~7 in the revision, we added a new comparison of Super-Linear (full-shot) to recent LLM- and VLM-based forecasting models, including Time-LLM, Time-VLM, and TimeCMA. Super-Linear achieves the best results in 8 MSE and 10 MAE configurations, competitive with Time-LLM (10 MSE, 8 MAE), despite using only 2.5M parameters compared to 143M (Time-VLM) and 3,405M (Time-LLM). Moreover, Super-Linear is trained once on the 96-step horizon and evaluated on longer horizons (192, 336, 720) without retraining, whereas competing methods train separate models per dataset and horizon. Finally, we follow a strict validation-based hyperparameter selection protocol, while prior LLM- and VLM-based works report varying configurations without a clearly defined validation-driven search procedure. Overall, Super-Linear delivers competitive performance with significantly greater efficiency and methodological consistency.
>
> We added below the average rows of each dataset. The full can be found in the revision.
>
>
> | Dataset | Super-Linear MSE | Super-Linear MAE | TimeCMA MSE | TimeCMA MAE | Time-VLM MSE | Time-VLM MAE | Time-LLM MSE | Time-LLM MAE |
> |---|---|---|---|---|---|---|---|---|
> | ETTh1 | 0.402 | 0.420 | 0.427 | 0.430 | 0.405 | 0.420 | 0.408 | 0.424 |
> | ETTh2 | 0.346 | 0.388 | 0.368 | 0.396 | 0.340 | 0.391 | 0.334 | 0.383 |
> | ETTm1 | 0.354 | 0.378 | 0.380 | 0.392 | 0.351 | 0.376 | 0.329 | 0.372 |
> | ETTm2 | 0.255 | 0.310 | 0.275 | 0.323 | 0.248 | 0.311 | 0.251 | 0.314 |
> | Electricity | 0.162 | 0.254 | 0.173 | 0.268 | 0.172 | 0.272 | 0.159 | 0.253 |
> | Weather | 0.221 | 0.259 | 0.250 | 0.276 | 0.224 | 0.263 | 0.225 | 0.258 |
>
> ---
>
> >It would be better to include a case study to intuitively show the effectiveness of the proposals.....
>
>  Thanks for this remark. We added a case study of various datasets, comparing super-linear to Time-Moe, Timer-XL, and Sundial in the appendix of the revision section I (last section).

---

### Review · Reviewer_pMh2 · 2026-02-23

**Summary Of Contributions:**

### Summary Of Contributions

The paper introduces Super-Linear, a time series forecasting model based on a mixture of linear experts. The approach follows a two-stage training procedure. In the first stage, each expert is pretrained on data corresponding to a specific temporal frequency. In the second stage, a gating mechanism is trained to combine these frequency-specialized linear models.
The proposed framework targets zero-shot forecasting in the univariate setting. By leveraging frequency-specialized linear components and a lightweight aggregation mechanism, the method aims to provide a computationally efficient alternative to large deep learning foundation models, while maintaining competitive performance.


### Strengths And Weaknesses:

#### Strengths

* S1.  The paper is well written and the contributions are clearly stated. Although the method relies on design choices (e.g., number of experts, predefined frequencies), the overall approach is coherent and well motivated.
* S2. While much of the recent literature has focused on increasingly large and computationally intensive time series foundation models (TSFMs), lightweight alternatives remain relatively underexplored. This direction is particularly relevant for real-world applications where computational resources are limited, and the paper makes a meaningful contribution in this space.
* S3. Compared to the selected baselines, the results on the LTSF benchmark are strong. The inference time comparison is clearly presented and highlights a practical advantage of Super-Linear in terms of computational efficiency. The results on the Gift-Eval benchmark are also informative, as they illustrate the limitations of purely linear forecasters in more challenging settings.
* S4. The section on forecasting under different sampling rates is particularly interesting. The associated discussion sheds light on potential frequency biases in current TSFM benchmarks and contributes to a broader reflection on evaluation practices in the literature.

#### Weaknesses

* W1. Although the paper is generally well written, a few sentences would benefit from clarification. For example, in the introduction, the statement “whereas Chronos can take over 4 seconds to generate an output for a batch size of 64 under certain configurations” should specify what is meant by “certain configurations.” Providing more precise experimental conditions would improve transparency.
* W2.  A key property of TSFMs is not only strong zero-shot performance but also flexibility, i.e., the ability to handle varying look-back window lengths and produce forecasts for arbitrary horizons. Super-Linear addresses this through up/downsampling of the look-back window and roll-out forecasting for horizons longer than 96 timesteps. However, these design choices may affect performance. A more detailed discussion of the potential trade-offs and limitations would strengthen the paper.
* W3. Recent TSFMs (e.g., TabPFN-TS and Chronos-2) allow the inclusion of covariates at inference time. It would be valuable to discuss whether and how Super-Linear could be extended to incorporate exogenous variables, and what architectural or training modifications this would require.
* W4. While it is understandable that not all existing TSFMs can be included in the experimental comparison, the absence of a comparison with the recent Chronos-2 model is noticeable. Including this baseline would likely strengthen the empirical evaluation and improve the paper’s positioning within the current literature.

**Additional Comments:**

No

**Audience:**

Yes

**Audience Explanation:**

As stated above, while much of the recent literature has focused on increasingly large and computationally intensive time series foundation models (TSFMs), lightweight alternatives remain relatively underexplored. This direction is particularly relevant for real-world applications where computational resources are limited, and the paper makes a meaningful contribution in this space.

**Claims And Evidence:**

Yes

**Claims Explanation:**

The main claims are generally supported by empirical evidence. The experimental evaluation covers multiple benchmarks, includes comparisons with relevant baselines, and provides both accuracy and inference time analyses. The results consistently support the claim that Super-Linear offers competitive zero-shot performance while being significantly more computationally efficient than larger TSFMs.
While some additional comparisons (e.g., with more recent baselines) and discussions would further strengthen the evidence, the current experimental results are sufficiently clear and convincing to support the core claims of the paper.

**Requested Changes:**

Please see the above weaknesses.

---

> ### Author Response · Authors · 2026-03-08
>
> We sincerely thank the reviewer for the positive and thoughtful evaluation of our work. We greatly appreciate the recognition of the clarity of the writing, the relevance of lightweight TSFMs, and the thorough experimental analysis, particularly regarding efficiency and sampling-rate robustness. In the revision, we will clarify the experimental configurations (e.g., Chronos timing), expand the discussion on look-back resampling and roll-out trade-offs, address the integration of covariates, and strengthen the positioning with respect to recent TSFMs such as Chronos-2.
>
> ---
>
> >W1. Although the paper is generally well written, a few sentences would benefit from clarification. For example, in the introduction, the statement “whereas Chronos can take over 4 seconds to generate an output for a batch size of 64 under certain configurations” should specify what is meant by “certain configurations.” Providing more precise experimental conditions would improve transparency.
>
> Thank you for this helpful comment. The experimental setup corresponding to this statement is described in Section~4.3. Specifically, the reported runtime refers to the configuration used in our main efficiency comparison: batch size 64, forecasting horizon 32, and lookback window 512.
>
> To improve clarity and transparency, we revised the sentence in the introduction by replacing the phrase ``under certain configurations'' with a precise reference to the experimental setup in Section~4.3. This change makes the conditions under which the reported runtime was measured explicit.
>
> ---
>
> >W4. While it is understandable that not all existing TSFMs can be included in the experimental comparison, the absence of a comparison with the recent Chronos-2 model is noticeable. Including this baseline would likely strengthen the empirical evaluation and improve the paper’s positioning within the current literature.
>
> In the revision, we have added Chronos-2 to
>
> (i) the performance–vs.–inference trade-off analysis in Figure~1,
>
> (ii) the GIFT-Eval comparison in Figure~5, and
>
> (iii) the inference time comparison in Section~4.3.
>
> While Chronos-2 is a strong and competitive baseline, Super-Linear remains more than 40$\times$ faster at inference and is substantially more parameter-efficient (2.5M vs.\ 120M parameters), highlighting its favorable efficiency–performance trade-off.

---

> > ### Author Response · Authors · 2026-03-08
> >
> > >W2. A key property of TSFMs is not only strong zero-shot performance but also flexibility, i.e., the ability to handle varying look-back window lengths and produce forecasts for arbitrary horizons. Super-Linear addresses this through up/downsampling of the look-back window and roll-out forecasting for horizons longer than 96 timesteps. However, these design choices may affect performance. A more detailed discussion of the potential trade-offs and limitations would strengthen the paper.
> >
> > Thanks for this remark, we added a new section in revision's appendix to discuss the method's flexibility tradeoffs, also provided here:
> >
> > A defining property of TSFMs is flexibility across varying look-back lengths and forecast horizons. Super-Linear achieves this via
> > (i) resampling-based alignment of the input window to the fixed training length $L_{\text{train}}=512$, and
> > (ii) autoregressive roll-out for long horizons.
> > While effective, these mechanisms introduce controlled but non-negligible trade-offs.
> >
> > ### Varying Look-back Lengths
> >
> > #### $L_{\text{input}} < 512$
> >
> > Short inputs are upsampled via linear interpolation and the forecast is subsequently downsampled back to the original resolution.
> >
> > **Benefits**
> >
> > - Ensures compatibility with the trained frequency experts while preserving dominant low-frequency structure.
> >
> > **Limitations**
> >
> > - *No new information*: Upsampling increases resolution without adding spectral content.
> > - *Smoothing bias*: Linear interpolation attenuates high-frequency components.
> > - *Spectral distortion*: Interpolation may alter energy distribution and increase the residual energy outside the expert span $\mathcal{S}$.
> >
> > From the bound
> >
> > $$
> > \|Y(X)-\hat{Y}(X)\|
> > \le
> > \sqrt{E_{\perp}(X)}
> > +
> > \gamma \|X\|_2 \|\beta-G\|_1,
> > $$
> >
> > interpolation can increase $E_{\perp}(X)$ if spectral mass is shifted outside $\mathcal{S}$. Moreover, very short look-backs limit observability of long-period components, constraining achievable approximation quality irrespective of resampling.
> >
> > #### $L_{\text{input}} > 512$
> >
> > Long inputs are adaptively downsampled by minimizing
> >
> > $$
> > \text{score}=
> > \text{entropy}(G(X_{\text{interp}}))
> > +
> > \lambda A_{\text{penalty}}[s],
> > $$
> >
> > subject to an energy-loss constraint.
> >
> > **Benefits**
> >
> > - Aligns dominant frequencies with the expert span and explicitly controls energy removal.
> >
> > **Limitations**
> >
> > - *Information loss*: Frequencies above the induced Nyquist limit are discarded,
> >
> > $$
> > E_{\text{lost}}=\sum_{\omega_j>\omega_{\text{Nyq}}} I_j.
> > $$
> >
> > - *Span limitation*: If dominant components lie outside $\mathcal{S}$, the irreducible term $\sqrt{E_{\perp}(X)}$ remains.
> >
> > ### Arbitrary Horizons via Roll-Out
> >
> > For horizons exceeding the native prediction length, Super-Linear applies autoregressive roll-out.
> >
> > **Benefits**
> >
> > - Enables arbitrary forecast lengths without architectural modification.
> >
> > **Limitations**
> >
> > - *Error accumulation*: Estimation error propagates across steps.
> > - *Additional overhead*: Long horizons require multiple forward passes due to roll-out. Nevertheless, Super-Linear remains highly efficient, and overall inference time is still significantly lower than attention-based TSFMs.
> > - *Amplified gating deviation*: The estimation term
> >
> > $$
> > \gamma \|X\|_2 \|\beta-G\|_1
> > $$
> >
> > may accumulate over long horizons.
> >
> > ### Architectural Trade-off
> >
> > Super-Linear achieves flexibility through *spectral adaptation* rather than architectural expansion. This provides robustness to unseen sampling rates and controlled frequency alignment, but expressivity remains bounded by the predefined expert set $\mathcal{S}$. Unlike large attention-based TSFMs, new frequency bases are not dynamically constructed at inference time.
> >
> > ### Summary
> >
> > Flexibility in Super-Linear is achieved through controlled resampling and autoregression. These mechanisms preserve robustness and interpretability but introduce smoothing bias, potential energy loss, and roll-out error accumulation. The theoretical bound highlights that performance ultimately depends on
> > (i) residual spectral energy outside $\mathcal{S}$ and
> > (ii) gating estimation error,
> > both of which define the model’s intrinsic flexibility limits.

---

> > > ### Comment · Reviewer_pMh2 · 2026-03-09
> > > **Answer to authors**
> > >
> > > Thanks for answering my questions. I don’t have any more concerns.

---

### Review · Reviewer_16zx · 2026-02-27

**Summary Of Contributions:**

This work combines two ideas into one: linear experts are shown to have good performance for time series forecasting and MoE model is also used in other foundation time series models. It replaces deep neural architectures with a set of simple linear experts, each specialized for different frequency patterns and trained on data resampled across multiple frequency regimes. A lightweight spectral gating module dynamically selects the most relevant experts, Despite its simplicity, Super-Linear achieves good benchmark performance while significantly enhancing efficiency, robustness to varying sampling rates, and interpretability due to the use of linear model. I appreciate the authors’ effort on the theoretical analysis of the gating mechanism.

**Audience:**

Yes

**Audience Explanation:**

Time series foundation model is an active developing research field.

**Broader Impact Concerns:**

N/A.

**Claims And Evidence:**

No

**Claims Explanation:**

The paper does not have sufficient clarity in my opinion. Hence, it's hard to assess the correctness. I have following questions about the proposed method:
* “If the lookback Linput is shorter than the training length (Ltrain = 512), the series is upsampled via linear interpolation; if longer, it is downsampled while preserving signal energy.” How exactly these are done?
* When time series is shorter, why not support padding from the left with missing values? When it’s longer, why down-sample? I have seen many work just keeps the last L values. This sampling may change the forecasting problems in the benchmarks and has an unknown impact.
* More importantly, in the evaluation of Table 1, are the up and down sampling also applied to the test set? Taking results from other papers directly do not make sense if such sampling is applied only on the proposed method.
* How does gaiting function different than the Time-MoE paper? This should be put into context when introducing gating function in Section 3. If different, why not just using the gating from Time-MoE? Is it a better thing to do in general?
* Any overlap between pre-training datasets (Monash and PEMS) and testing datasets (GIFT eval)? This potentially has a big impact.
* There is also pre-training dataset resampling (in Appendix D3). I want to know with and without it, how does it change the results.

**Requested Changes:**

I hope to get answers for the questions I raised above, those are critical.

And I think the writing needs to be improved in general. I point out some places as examples:
* Better to explain what is full-shot setting in the very beginning of the paper.
* What does $I(X-\hat{X})$ represent? More explanation would be nice to have.
* What does it mean in Training Mechanism section “that provide sufficient sequence length for downsampling”?
* In the problem formulation section, why channel independence enables multivariate forecasting if the time series are modeled independently?
* What does “Lookback weights” and “lookback adaptability” mean?
* “…we transform each dataset to span a wider range of frequencies, as illustrated in 11 and listed in 11…” This is redundant wording.

---

> ### Author Response · Authors · 2026-03-08
>
> We thank the reviewer for the careful reading and constructive feedback. We appreciate the comments on the motivation, efficiency, and theoretical aspects of our method. Below we address the reviewer’s questions.
>
> ---
>
> >If the lookback Linput is shorter than the training length (Ltrain = 512), the series is upsampled via linear interpolation; if longer, it is downsampled while preserving signal energy.” How exactly these are done?
>
> We provide full details in Section B.8, including a detailed algorithm. Briefly, Super-Linear handles variable lookback lengths through simple resampling.
>
> ### **L_input < 512 (Upsampling)**
>
> When the input lookback is shorter than the training length
> `L_train = 512`, we:
>
> - Upsample the series via linear interpolation to length `512`
> - Perform inference at this resolution
> - Downsample the forecast back to the original resolution
>
> The resampling factor is defined as:
>
>
> $s = \left\lceil \frac{512}{L_{\text{input}}} \right\rceil$
>
> After prediction, we interpolate back using a factor of `1/s`.
> Although interpolation modifies the effective frequency content,
> Super-Linear remains robust due to its frequency-specialized experts.
>
>
> ### **L_input > 512 (Adaptive Downsampling)**
>
> For longer lookbacks, we adaptively downsample using a set of
> candidate scales (default `S = [2,4,6]`). For each scale `s ∈ S`, we:
>
> 1. Estimate signal energy loss using the normalized periodogram (PSD)
> 2. Compute the entropy of the expert gating distribution
> 3. Apply linear interpolation to resample the series
>
> We select the scale minimizing:
>
> $ \text{score}(s) =\text{entropy}(G(X_s)) + \lambda \cdot \text{energy-penalty}(s) $
>
> This objective balances preserving signal energy ($\text{energy-penalty}(s) $) and maintaining
> confident expert selection ($\text{entropy}(G(X_s))$). The chosen scale is used both for input
> resampling and for restoring the forecast to the original granularity.
>
>
> In summary, short sequences are linearly upsampled, and long sequences
> are energy-aware adaptively downsampled, ensuring compatibility with
> the fixed training length while preserving frequency structure.
>
> ---
>
> >When time series is shorter, why not support padding from the left with missing values? When it’s longer, why down-sample? I have seen many work just keeps the last L values. This sampling may change the forecasting problems in the benchmarks and has an unknown impact.
>
>
> ### Why not left-padding when `L_input < 512`?
>
> Consider a linear forecaster with weights
> $W \in ℝ^{512}$ and input $X \in ℝ^{512}$.
> If a shorter sequence is left-padded (e.g., with zeros), the padded input becomes $X₀$, where a prefix of the vector is zero.
>
> Since the model is linear, the prediction is:
>
>
> $\hat{y} = X W $
>
>
> If padding is introduced at inference time but the model was trained without padding, then:
>
>
>  $ | X_0 W | < | X W | $ ,
>
>
> because part of the weight vector multiplies zeros. This effectively removes contributions from a subset of learned coefficients, even if those weights are important for capturing temporal structure.
>
> Unlike deep architectures, a purely linear model has no additional layers to **compensate** for this lost signal energy. As a result, predictions become unintentionally biased toward zero.
>
> Conversely, if the model were trained with padding, it would learn to suppress weights corresponding to typical padded regions, reducing the model’s effective capacity.
>
> To avoid wasting model capacity and distorting the learned weight structure, we instead **upsample shorter sequences via interpolation**.
>
>
> ### Why downsample when `L_input > 512` instead of keeping the last 512 values?
>
> Keeping only the last 512 observations is equivalent to **cropping a high-resolution signal**. This may discard substantial long-range structure present in sequences of length `2000–4000`.
>
> An alternative is to **shrink the resolution while preserving signal energy**, analogous to resizing an image rather than cropping it. Our adaptive downsampling preserves global frequency content and selects the scale that minimizes energy loss and gating entropy (Section B.8).
>
> Empirically, this procedure is important. In **Table 2**, GIFT-Eval without resampling performs worse (`MASE = 0.92`) compared to the energy-aware downsampling variant (`MASE = 0.857`), demonstrating that naive cropping degrades forecasting accuracy.
>
> ---
>
> >More importantly, in the evaluation of Table 1, are the up and down sampling also applied to the test set? Taking results from other papers directly do not make sense if such sampling is applied only on the proposed method.
>
> No. In Table~1, the lookback length is fixed to $512$ for all methods to ensure a fair comparison with prior work. No resampling is applied there.
>
> In GIFT-Eval, lookback lengths vary widely (from dozens to thousands), so we apply the proposed resampling strategy.
>
> ---

---

> > ### Author Response · Authors · 2026-03-08
> >
> > >How does gaiting function different than the Time-MoE paper? This should be put into context when introducing gating function in Section 3. If different, why not just using the gating from Time-MoE? Is it a better thing to do in general?
> >
> >
> > The gating mechanisms in Time-MoE and Super-Linear are fundamentally different.
> >
> > Super-Linear routes experts using explicit frequency information extracted via FFT, with each expert specialized to a specific frequency regime. The router operates directly on spectral features of the raw signal.
> >
> > In contrast, Time-MoE routes based on hidden token embeddings within a deep architecture and does not explicitly model frequency structure. Its experts are generic rather than frequency-specialized.
> >
> >
> > Therefore, Time-MoE’s gating is not directly compatible with Super-Linear’s linear, frequency-specialized design.
> >
> > ---
> >
> > >Any overlap between pre-training datasets (Monash and PEMS) and testing datasets (GIFT eval)? This potentially has a big impact
> >
> > There is limited overlap: the Saugeen, Births, and KDD datasets appear in both the pretraining corpus and GIFT-Eval. Regarding the Solar dataset, it was not included in the training corpus; it was mistakenly added to the training data table in Table 12 in the original manuscript; however was removed in the revision. Nevertheless, several competing methods (e.g., Moirai, TTM-R1/R2, Time-MoE, Chronos-Bolt/Base, TimesFM 1.0/2.0, and likely Sundial) are trained on LOTSA [1] or parts of it, which also contains datasets used in GIFT-Eval. This is important context for fair comparison.
> >
> > To quantify the impact, we trained a version of Super-Linear excluding the overlapping datasets (“no leak”). Results are shown below.
> >
> > ### LTSF Results (MASE)
> >
> > | Model | ETTm1 | ETTm2 | ETTh2 | ETTh1 | Traffic | Weather | Electricity |
> > |---|---|---|---|---|---|---|---|
> > | Super-Linear | 0.317 | 0.179 | 0.279 | 0.369 | 0.414 | 0.159 | 0.141 |
> > | Super-Linear (no leak) | 0.316 | 0.177 | 0.277 | 0.371 | 0.412 | 0.162 | 0.148 |
> >
> > ### GIFT-Eval Results (MASE)
> >
> > | Model | GIFT-Eval (MASE) |
> > |---|---|
> > | Super-Linear | 0.857 |
> > | Super-Linear (no leak) | 0.863 |
> >
> > The differences are marginal and mixed, indicating that overlap has minimal impact. This further supports that **Super-Linear relies primarily on frequency diversity rather than sheer data abundance**.
> >
> > ---
> > # Clarifications:
> >
> > >Better to explain what is full-shot setting in the very beginning of the paper.
> >
> > Thanks for this remark, we added an explanation as part of the introduction in the revision.
> >
> >
> > >What does $I = (X - \bar{X})$ represent? More explanation would be nice to have
> >
> >
> > this represents the normalized periodogram $I(X - \bar{X})$ centered zero-mean input.
> > We emphasized the notation and improved clarity in the revision. Nevertheless, the expression is given in Sec 3. in the "Gating Function" paragraph.
> >
> >
> > >What does it mean in Training Mechanism section “that provide sufficient sequence length for downsampling”?
> >
> > Our training corpus consists of datasets containing channels with sequence lengths exceeding 5,000 time steps. This is essential because our training procedure relies heavily on downsampling (and upsampling) to expose the model to a diverse range of frequencies. We discuss this in detail in Section C of the appendix; however, to improve clarity, we will incorporate this explanation into the main text as well.
> >
> >
> > >In the problem formulation section, why channel independence enables multivariate forecasting if the time series are modeled independently?
> >
> > This is a common practice for multivariate forecasting; instead of modeling channels together, potentially to make use of cross-channel information, channel independence addresses each channel separately in a multivariate setting.
> > This approach was also employed in Time-Moe, Timer-XL, Chronos-bolt, TiRex, and many others.
> >
> >
> > >What does “Lookback weights” and “lookback adaptability” mean?
> >
> > The lookback weights mean the shape of the model weights, which are fixed on the training lookback, namely $L = 512$, where $L$ represents the lookback.
> >
> > given a basic linear form: $   X W_i + b_i  $. In our work the weight's shape is fixed to $W_i \in \mathbb{R}^{L \times H}$ and $b_i \in \mathbb{R}^H$, with $H$ representing the forecast horizon.
> >
> > The paragraph in the paper that you're referring addresses the **Varying Lookbacks Adaptability**, cases where the input $X$ is with lookback $L\ne512$. Consequently, this paragraph discusses how Super-Linear can **adapt** to varying lookbacks.
> >
> > We also noticed that there may have been a notation confusion between $T$ and $L$ in the current version. We will fix the **lookback* notation to $L$ in the revision.
> >
> >
> > >“…we transform each dataset to span a wider range of frequencies, as illustrated in 11 and listed in 11…” This is redundant wording.
> >
> > Thanks for this remark, we fixed this phrase in the revision.
> >
> >
> > [1] Unified Training of Universal Time Series Forecasting Transformers

---

> > ### Comment · Reviewer_16zx · 2026-03-10
> > **Further comments on up-and-down sampling**
> >
> > I don't think the proposed upsampling and downsampling are standard things to do. This is my first time seeing this. It's a very big design change.
> >
> > When I say left-padding, I mean padding with nan, not zero. That's a standard thing to do. During training, depending on the training series length, the nan-padding may be present or not. This is how the model learn to handle series of different lengths during inference time.
> >
> > When you down sample, instead of taking the last 512 values, you access to more history information. In the GIFT evals, are the other methods fed into the same input? Probably not, the authors wrote "Note that theGIFT-Eval leader-
> > board is continuously updated, and our results reflect peer-reviewed models available at the time stated in Fig. 5.". If the input of the proposed method have access to much longer history through down sampling, is it still a fair comparison?

---

> > > ### Author Response · Authors · 2026-03-14
> > >
> > > > I don't think the proposed upsampling and downsampling are standard things to do. This is my first time seeing this. It's a very big design change.
> > >
> > > Thank you for raising this point. While the use of upsampling and downsampling in this context may not yet be common practice, we view it as a novel design choice aimed at enabling the model to better handle varying temporal resolutions within a unified framework.
> > >
> > >
> > > > When I say left-padding, I mean padding with nan, not zero. That's a standard thing to do. During training, depending on the training series length, the nan-padding may be present or not. This is how the model learn to handle series of different lengths during inference time.
> > >
> > > Thank you for the clarification. We agree that NaN padding is commonly used in many time-series models to handle variable-length inputs. However, in practice NaNs cannot be directly processed and are typically **converted to zeros and handled via masking**, as done in models such as Sundial and TiRex. Other approaches alternatively **truncate the signal** to a fixed length.
> > >
> > > However, this strategy is **not directly applicable to Super-Linear due to its linear backbone**. In a purely linear architecture, padding with NaNs (which are converted to zeros) effectively removes the contribution of the corresponding weights. As a result, part of the linear weight matrix becomes unused, reducing the effective weight energy participating in the prediction. In practice, this biases the output toward smaller magnitudes (closer to zero), which degrades the model's behavior, as explained in our previous reply.
> > >
> > > Handling missing or padded regions properly typically requires **additional nonlinear modeling capacity**, such as MLP layers with nonlinear activations that can learn to compensate for masked inputs. This is indeed the strategy used in models like Sundial, TiRex, and other deep architectures.
> > >
> > > Because Super-Linear intentionally relies on a **simple linear backbone**, such mechanisms are not available. Therefore, we instead adopt **signal resampling** so that the input matches the required linear weight dimension while preserving the information content of the signal.
> > >
> > > In summary:
> > >
> > > - **NaN padding** → linear layers cannot process NaNs directly (they must be converted and masked).
> > > - **Zero padding** → part of the linear weights becomes unused, biasing predictions toward zero.
> > > - **Truncation** → potentially important information from the signal is discarded.
> > >
> > > For these reasons, Super-Linear uses **resampling as a simple and consistent way to adapt variable-length signals to the fixed linear weight size**, while avoiding the issues above.
> > >
> > >
> > > > When you down sample, instead of taking the last 512 values, you access to more history information. In the GIFT evals, are the other methods fed into the same input? Probably not,
> > >
> > > Thank you for raising this point.
> > >
> > > GIFT-Eval consists of **more than 90 different dataset configurations**. Each dataset contains multiple forecasting examples of the form $(x, y)$, where the **input length $|x|$ can vary significantly**, ranging from only a few dozen timesteps to several thousand. In many cases, the available history can therefore be substantially longer than the minimal context required by a model.
> > >
> > > Importantly, the models evaluated in GIFT-Eval **do not use a unified input context length**. Instead, each method operates with its own **maximum context window**, allowing it to utilize as much historical information as its architecture permits. For example:
> > >
> > > - **Sundial** uses a maximum context length of **2880**
> > > - **TimesFM** uses **2048**
> > > - **Moirai** uses **4000**
> > > - **Super-Linear** uses **3072**
> > >
> > > It is also likely that models such as **TiRex** and **VisionTS** similarly operate with relatively long maximum contexts.
> > >
> > > Therefore, when longer input histories are available in the benchmark, **different models may leverage different amounts of historical information depending on their maximum supported context length**. In this sense, Super-Linear's use of resampling to fit its context window is consistent with the general evaluation setup, where each model processes the available history according to its architectural constraints.
> > >
> > >
> > > > the authors wrote "Note that theGIFT-Eval leader- board is continuously updated, and our results reflect peer-reviewed ...."
> > >
> > > Thanks for noting this, we will update the date in the revision.
> > >
> > > > If the input of the proposed method have access to much longer history through down sampling, is it still a fair comparison?
> > >
> > > Please see our response above for details.
> > >
> > > In short, the comparison remains fair because **models in GIFT-Eval already use different maximum context lengths**, allowing them to leverage different amounts of history depending on their architecture (see Figure 5).
> > >
> > > For the **LTSF benchmark (Table 1)**, most prior work uses an **input length of 512 or similar**, so we **also fix the input length to 512** for Super-Linear to ensure a fair comparison.

---

> > > > ### Comment · Reviewer_16zx · 2026-03-15
> > > > **Reply to authors**
> > > >
> > > > >However, this strategy is not directly applicable to Super-Linear due to its linear backbone
> > > >
> > > > I acknowledge due to the linear backbone, nan can't be used as input.
> > > >
> > > > >Importantly, the models evaluated in GIFT-Eval do not use a unified input context length. Instead, each method operates with its own maximum context window
> > > >
> > > > Thank you for pointing this out. I missed this point. It's a shame that we have to do that. The proposed method does not have a context window concept on the input and it can work with any length (due to the usage of sampling) as long as there is more than 1 data point in the input (correct me if I'm wrong). This should be highlighted in the paper.
> > > >
> > > > Then how did you do the ablations in the Table 2 when no sampling is used in the training? If no sampling is used, how did you handle training time series with various length? Assuming there is a principled way, should this then be the main results so that we can focus on the architecture change proposed by the paper, rather than this design change on the input?

---

> ### Author Response · Authors · 2026-03-18
>
> > Thank you for pointing this out. I missed this point. It's a shame that we have to do that. The proposed method does not have a context window concept on the input and it can work with any length (due to the usage of sampling) as long as there is more than 1 data point in the input (correct me if I'm wrong). This should be highlighted in the paper.
>
> Thank you for the suggestion. We have clarified in Section 3 (Varying Lookbacks Adaptability) that the model supports arbitrary input lengths, as long as the input includes at least two data points. This was added to the most recent revision.
>
> > Then how did you do the ablations in the Table 2 when no sampling is used in the training? If no sampling is used, how did you handle training time series with various length? Assuming there is a principled way, should this then be the main results so that we can focus on the architecture change proposed by the paper, rather than this design change on the input?
>
> Super-Linear is not trained on time series of varying lengths. As described in Section 4.1 (Training and Implementation), the model is trained using a fixed lookback length of 512. In our training pipeline, all training examples are resampled (when needed) to ensure a consistent length of 512. Accordingly, we select datasets with sufficiently long sequences (e.g., length ≥ 5,000), as also stated in Section 4.1.
>
> For the aforementioned ablation study in Table 2, we modify this procedure. Specifically, we use the raw signal and directly extract training examples of length 512, without applying the resampling step (e.g., we do not upsample shorter sequences such as length 96 to 512). In addition, we skip Stage 1 of training and instead train the entire model end-to-end in a single stage, without separating expert training and router training.
>
>
>
> We hope these clarifications and the clarifications above resolve the concerns, and we would be happy to provide further details if any additional questions arise.

---

> > ### Comment · Reviewer_16zx · 2026-03-19
> > **The position of the paper**
> >
> > >Accordingly, we select datasets with sufficiently long sequences (e.g., length ≥ 5,000), as also stated in Section 4.1.
> >
> > This is why in the ablation, the model can work. Because there is no series of length less than 512.
> >
> > I want to re-state my main point: The paper proposed for mixture of linear experts, that is a architecture change. This change is combined with a sampling procedure on the input, which has a big impact (as shown in the ablation). I think the main results should be without using the sampling (as in ablation) to highlight the architecture change unless the message of the paper change. When the training sampling is not used, how does the proposed model compared with other methods? Can the authors give an overview of that results?

---

> ### Author Response · Authors · 2026-03-19
>
> Thanks again for engaging in the discussion.
>
> We would like to emphasize that, while the proposed architecture is unique, it is fundamentally intertwined with the resampling procedure used during training. Evaluating the architecture in isolation, without the resampling component, would not faithfully represent the core contribution or the intended message of our work.
>
> As stated in the introduction of the manuscript:
>
> > *“We introduce Super-Linear, a sparse mixture-of-experts (MoE) model that brings modularity and frequency-awareness to linear forecasting. Super-Linear comprises linear heads, most of which are trained to specialize in a particular frequency regime, using resampled data to ensure abundance and diversity of periodic signals.”*
>
> A key aspect of Super-Linear is its reliance on **frequency abundance**, which is achieved through the resampling strategy. This is critical for two main reasons:
>
> 1. **Expert specialization (Stage 1 training):**
>    The resampling process ensures that each linear expert is exposed to sufficient examples of specific frequency regimes, allowing it to specialize effectively.
>
> 2. **Router learning  (Stage 2 training):**
>    The routing mechanism requires a diverse distribution of frequencies in order to learn how to correctly assign inputs to the appropriate experts based on their frequency characteristics.
>
> Without resampling in training:
> - There is insufficient diversity and representation of different frequency regimes in the training data.
> - Training directly on the raw dataset is likely to result in *frequency confusion*, where multiple frequency patterns compete within a single linear model. This phenomenon is discussed in Section 3 (Motivation) and illustrated in Figure 2, where overlapping frequency components interfere with the learned weight structure.
>
> It is also worth noting that different models rely on different data augmentation and training regimes. For example, Chronos leverages synthetic data along with augmentation techniques such as Mixup, while TiRex augments the data with “spiky” seasonal patterns. More broadly, many approaches rely on synthetic data to enrich the training corpus. In contrast, Super-Linear does not use synthetic data; instead, it employs resampling as a mechanism to increase frequency diversity and ensure adequate coverage of different periodic regimes.
>
> For completeness, we note that the full ablation results from Table 2 are further detailed in Appendix B.8 (including MAE and MSE without resampling).
>
> However, presenting these ablation results as the primary comparison would not align with the central motivation of our work—namely, improving linear forecasting through **frequency-aware modularity**. The architecture and the resampling procedure are designed to work jointly, and isolating one from the other would obscure the intended contribution.

---

> > ### Comment · Reviewer_16zx · 2026-03-19
> > **Confusion between resampling on dataset and resampling on the input**
> >
> > I am confused by the term "resampling" in the paper. There are two, one resampling for dataset augmentation and one for making sure the input series is 512 long before fed into the model. I wanted to talk about the 2nd resampling in this whole discussion. The author wrote "Lastly, training only on the raw datasets without resampling during training ..." and I thought that is for the 2nd resampling, to make sure the input length is 512. It seems, what the authors meant is the first resampling for dataset augmentation. I think the data augmentation part is fine.
> >
> > It is important to know the impact of resampling on the input, that's a big design change and it is under-stated in this paper.  The key question for me is that how much of the good performance that Superlinear get is due to this under-stated but potentially impactful change.
> >
> > It's still possible to do something in this direction. For example, one could filter training datasets with time series at last 512 data points long and compare two strategies: A) using the proposed down-sampling and B) using only the last 512 data points. They are useful for two reasons:
> > - When compare these two strategies using the proposed model and training, one can know the benefit of using down-sampling when combined with proposed model and training.
> > - When using last 512 datapoints (B) and train other baselines and the proposed model on the same input, one can know how does the proposed model compare with other model.
> >
> > A light weight test can be feeding the baselines with the same input (after resampling to 512) as SuperLinear and then compare the performance.

---

> > > ### Author Response · Authors · 2026-03-19
> > >
> > > Thanks for clarifying your question, I believe I misunderstood your concern in the previous comment.
> > >
> > > To address this point, the ablation study in Table 2 includes a column labeled **“WO Inf Downsampling”** (previously “WO Resampling”). We renamed it in the revisioni for clarity to explicitly refer to **inference-time downsampling**.
> > >
> > > This ablation corresponds to the setting where the model is given only the **last ≤512 timesteps** of the series, *without* applying downsampling to incorporate longer history. In this setting, the reported MASE on the Gift-Eval benchmark is **0.92**, compared to 0.857 when downsampling is applied.
> > >
> > > It is important to note that, since Super-Linear expects a fixed input length, sequences shorter than 512 are still **resampled up to length 512**. However, no information beyond the most recent 512 timesteps is used — i.e., longer historical context is not incorporated via downsampling.
> > >
> > > Additionally, as mentioned earlier, all results reported in Table 1 correspond to a **lookback of 512** for Super-Linear.
> > >
> > > We agree that understanding the impact of downsampling is important, and this ablation is intended to isolate the effect of incorporating longer context via inference-time downsampling versus using only the most recent 512 observations.

---

> ### Comment · Reviewer_16zx · 2026-03-20
> **Final comments**
>
> I am glad the authors are moving into this direction. As the results show, the downsampling on the input has a large impact and making the proposed method degrade significantly on GIFT-eval (MASE: 0.857->0.92). And I can imagine when removing upsampling (not possible in the current setting but can be constructed), more performance drop will be observed. Another point that the authors have been called out, is the data augmentation, without which the method would be worse than Naive baseline (1.27 v.s. 1.603) on GIFT-eval. These results together suggest how to augment data and how to construct input are super-important. Actually they matter the most than the other architectural and training ablations in Table 2. I suggest the authors to explicitly call-out these observations in the abstract, introduction, ablation and the conclusion section.
>
> I'd like to also suggest the authors to make a pass on the manuscript again. There are a few places can be clearly improved. For example, in the description of Table 2, it's not common to explain the right table first and then the left table. In addition, the authors should explain why there are no results for WO Inf Downsamp column on the MSE and MAE row. Also, the MSE and MAE are for LTSF task, should the row names better be called LTSF-MSE and LTSF-MAE? These are just examples I noticed related to our discussion, and I believe there are more places can be improved.

---

> > ### Author Response · Authors · 2026-03-22
> >
> > Thank you for the constructive feedback. We have prepared a revised version of the manuscript that incorporates your suggestions.
> >
> > ### 1. Emphasis on Resampling and Data Augmentation
> >
> > We strengthened the discussion around the impact of resampling methods. In particular, we now highlight their importance in the abstract, introduction, and conclusion.
> > Additionally, Section 4.2 (Ablation Study) has been expanded to better emphasize the significance of both training-time augmentation and inference-time resampling, reflecting the observations you raised.
> >
> > ### 2. Improvements to Clarity and Presentation
> >
> > We performed another careful pass over the manuscript to improve clarity and readability throughout. Following your recommendations:
> >
> > The caption and description of Table 2 have been revised to follow a more conventional order.
> >
> > We clarified why the **WO Inf Downsamp** column does not include MSE/MAE results.
> >
> > We updated metric labels to explicitly reference the LTSF task (e.g., LTSF-MSE, LTSF-MAE).
> >
> > It is worth noting that several additional sections were already improved for clarity based on our previous discussion.
> >
> > Thank you again for your detailed comments, they helped us significantly improve the quality and polish of the paper.

---

### Author Response · Authors · 2026-03-08

We sincerely thank the reviewers and editors for their careful reading and constructive feedback, which helped improve the quality and clarity of our paper. In the revised version, we implemented the following changes:

- **Added comparison to Chronos-2** (Reviewer pMh2; Figure 1, Figure 5, and Table 3).
- **Added clarifications based on reviewer comments and fixed typos.**
- **In Table 2 (right), added a comparison to a setup that does not resample the data during training (WO Tr Resamp)** (Reviewer 16zx).
- **Added a new section comparing Super-Linear to LLM-based models** (Appendix B.4; Reviewer P2Up).
- **Added comparison to DUET and FEDformer** (Appendix Table 4; Reviewer P2Up).
- **Added a case study visualization** (Appendix I; Reviewer pMh2).
- **Added a section discussing trade-offs and limitations of Super-Linear’s flexibility** (Appendix B.11; Reviewer pMh2).

Text changes are marked in **red**, revised figures and tables have a **red caption**, and new sections have their title marked in **red**.

We look forward to engaging in further discussions and appreciate your continued feedback.

---

### Decision · Action_Editor_GeSL · 2026-04-16

**Recommendation:** Accept as is

**Additional Comments:**

Final decision: Accept as is.
Please ensure that the camera version of your paper is the revision incorporating all remarks from the discussion with Reviewer 16zx, in particular the requested ablation studies.

**Audience:**

Yes

**Audience Explanation:**

As all of the reviewers' noted, the time series foundation model is an active developing research field and there is TMLR audience interested in this topic.

**Claims And Evidence:**

Yes

**Claims Explanation:**

The reviewers highlighted the paper’s clear and convincing message, supported by experimental evidence, regarding the effectiveness of data augmentation, input adaptation, and lightweight alternatives to computationally intensive time-series foundation models.